# Regulation of mutant *TERT* by BRAF V600E/MAP kinase pathway through FOS/*GABP* in human cancer

Rengyun Liu [1], Tao Zhang[1], Guangwu Zhu[1] & Mingzhao Xing[1]

The unique oncogene duet of coexisting *BRAF* V600E and *TERT* promoter mutations are widely found to be a robust genetic background promoting human cancer aggressiveness, but the mechanism is unclear. Here, we demonstrate that the BRAF V600E/MAP kinase pathway phosphorylates and activates FOS, which in turn acts as a transcription factor to bind and activate the *GABPB* promoter, increasing GABPB expression and driving formation of GABPA-GABPB complex; the latter selectively binds and activates mutant *TERT* promoter, upregulating *TERT* expression. Elevated TERT functions as a strong oncoprotein, robustly promoting aggressive behaviors of cancer cells and tumor development. We thus identify a molecular mechanism for the activation of mutant *TERT* by the BRAF V600E/MAP kinase pathway, in which FOS as a transcriptional factor of *GABPB* promoter plays a key role in functionally bridging the two oncogenes in cooperatively promoting oncogenesis, providing important cancer biological and clinical implications.

[1] Laboratory for Cellular and Molecular Thyroid Research, Division of Endocrinology, Diabetes, & Metabolism, Department of Medicine, John Hopkins University School of Medicine, Baltimore, MD 21287, USA. Correspondence and requests for materials should be addressed to M.X. (email: mxing1@jhmi.edu)

Telomerase reverse transcriptase (TERT) is the catalytic component of the telomerase complex, which plays a key role in maintaining the telomere length of chromosomes and cell immortality and in controlling cellular activities[1]. Mutations in the *TERT* promoter were found initially in melanoma[2,3] and subsequently widely in other human cancers[4], including thyroid cancer[5,6]. Two recurrent *TERT* promoter mutations located at hotspots chr5, 1,295,228 C > T (C228T) and 1,295,250 C > T (C250T) are particularly common, which correspond to the positions 124 and 146 bp, respectively, upstream of the translation start site of *TERT*; both mutations are predicted to generate a consensus binding site for E-twenty-six (ETS) transcription factors[2,3]. Further studies showed that *TERT* promoter mutations were associated with higher levels of *TERT* expression, telomerase activities, and oncogenic cellular activities[7–10].

*BRAF* V600E is another major human oncogene that was also initially discovered in melanoma and subsequently widely found in other human cancers[11], including thyroid cancer, particularly papillary thyroid cancer (PTC)[12]. This mutation causes constitutive activation of the BRAF kinase and consequent oncogenic activation of the mitogen-activated protein kinase (MAPK) pathway through phosphorylating MEK and ERK. It has been widely observed that *BRAF* V600E is associated with aggressiveness of human cancer, as exemplified by increased tumor recurrence and disease-specific mortality of PTC[13,14] as well as clinicopathological aggressiveness of melanoma[15,16] and other cancers such as colorectal cancer and brain tumor[15,17,18]. *TERT* promoter mutations were similarly associated with increased aggressiveness of human cancers, as exemplified by increased tumor recurrence and disease-specific mortality of PTC[5] and aggressiveness of melanoma[9,19], glioma, and bladder cancer[20,21]. Interestingly, *BRAF* V600E was widely found to be associated with *TERT* promoter mutations in human cancers, particularly thyroid cancer and melanoma[2,5,6,9,19,22,23]. When separated from each other, either mutation alone had only a modest effect while coexisting *BRAF* V600E and *TERT* promoter mutations were associated with robustly increased cancer aggressiveness, as exemplified by increased lymph node metastasis, distant metastasis, advanced tumor stage, tumor recurrence, and disease-specific mortality of PTC[24–27]. In a recent large meta analysis on PTC[5], the prevalence of coexistence of *BRAF* V600E and *TERT* promoter mutations was 7.7% (145/1892), which impressively numerically corresponds to the conventionally known <10% of patients with PTC that has the poorest clinical outcomes. In melanoma, coexistence of *BRAF* V600E and *TERT* promoter mutations was associated with increased tumor thickness, high mitotic rate, lymph node metastasis, presence of ulceration, absence of regression, high risk of tumor recurrence, and melanoma-specific mortality[22,23].

These results establish that the unique oncogene duet of coexisting *BRAF* V600E and *TERT* promoter mutations is a fundamental genetic background that cooperatively drives progression and aggressiveness of some human cancers. However, the molecular mechanism underpinning the synergistic oncogenic operations of the two oncogenes is undefined. Specifically, a fundamental question as to how *BRAF* V600E and mutant *TERT*, which each apparently represents a different molecular system, are functionally bridged at the molecular level in cooperatively promoting human cancer aggressiveness remains to be answered. We mechanistically explored this issue in the present study by testing our hypothesis that the BRAF V600E-activated MAPK pathway may selectively upregulate the mutant *TERT* through a molecular mechanism that mediates the activation of the mutant *TERT* promoter-selective transcriptional machinery, thus functionally connecting the two oncogenes in cooperatively promoting oncogenesis. We identify FOS as playing such a critical role in this important mechanism of human oncogenesis. Specifically, we identify a molecular mechanism for the activation of mutant *TERT* by the BRAF V600E/MAP kinase pathway, in which FOS as a transcriptional factor of the *GABPB* promoter promotes the expression of GABPB; the latter complexes with GABPA and selectively binds and activates the mutant *TERT* promoter, robustly upregulating the expression of *TERT*, thus functionally bridging the two oncogenes in cooperatively promoting oncogenesis.

## Results

**Cooperative role of BRAF V600E and TERT in cell oncogenesis.** To support the clinical findings on the genetic duet of *BRAF* V600E and *TERT* promoter mutations and demonstrate its biological relevance, we used in vitro and in vivo models to examine the roles of BRAF V600E and TERT in oncogenic cellular activities and xenograft tumor development of PTC cells BCPAP and K1 and melanoma cells A375, which all harbored both *BRAF* V600E and *TERT* promoter mutations. As shown in Supplementary Fig. 1, BRAF shRNA effectively knocked down BRAF protein and suppressed ERK phosphorylation of the MAPK pathway; similarly, TERT siRNA knocked down more than 80% of TERT protein in the three cells. Knockdown of either BRAF or TERT significantly inhibited cell proliferation and dual knockdown of BRAF and TERT induced a further inhibition (Fig. 1a). Either BRAF or TERT knockdown decreased cell migration and invasion and dual knockdown of BRAF and TERT had more robust effects (Fig. 1b, c). Similarly, either BRAF or TERT knockdown inhibited anchorage-independent growth of K1 and A375 cells in soft agar and dual knockdown of BRAF and TERT nearly completely abolished colony growth (Fig. 1d). BCPAP cells naturally formed only a few colonies in soft agar and knockdown of either BRAF or TERT completely abolished the colony formation (Fig. 1d). We also tested the role of BRAF V600E and TERT in thyroid tumor growth using the BRAF V600E inhibitor PLX4032 and stable TERT knockdown to suppress the MAPK pathway and the TERT, respectively, in K1 cell (Fig. 1e), from which xenograft tumors were derived (Fig. 1f, g). Either administration of PLX4032 or TERT knockdown inhibited tumor growth and combination of the two nearly completely abolished tumor growth (Fig. 1f, g), particularly evident in tumor weight (Fig. 1g). These data demonstrated that, like BRAF V600E, TERT also played a robust role in cancer-hallmark oncogenic cellular activities and tumorigenesis of cancer cells; the oncogenic effect of TERT was in fact even more robust than BRAF V600E in these cells harboring both *BRAF* V600E and *TERT* promoter mutations. TERT was thus demonstrated here to be a powerful oncoprotein. These results also showed that BRAF V600E and TERT displayed a cooperative manner in promoting the oncogenic behaviors of these cells, recapitulating the clinical findings on the genetic duet of *BRAF* V600E and *TERT* promoter mutations.

**BRAF V600E and TERT mutation cooperatively upregulated TERT.** We next investigated whether *BRAF* V600E and *TERT* promoter mutations cooperatively affected *TERT* expression in a panel of human cancer cells with various *BRAF* and *TERT* genotypes (Supplementary Table 1). The *BRAF* V600E inhibitor PLX4032 specifically suppressed ERK phosphorylation in cells harboring *BRAF* V600E mutation and dramatically inhibited *TERT* expression in cells harboring both *BRAF* V600E and *TERT* promoter mutations, but had limited effect on *TERT* expression in cells harboring the wild-type *TERT* promoter and had no inhibitory effect in cells harboring wild-type *BRAF* (Fig. 2a, Supplementary Fig. 2a). In fact, several wild-type

*BRAF* cell lines showed an increase in ERK phosphorylation after the treatment with PLX4032, consistent with the previous finding that PLX4032 induced MEK and ERK phosphorylation in wild-type *BRAF* cells[28,29], correspondingly leading to an increase in TERT expression, especially in TERT mutant cells (KAT18, C643, and CHL-1) (Fig. 2a). The MEK inhibitor

AZD6244 similarly inhibited *TERT* expression in cells harboring *BRAF* V600E mutation, while it had a modest inhibitory effect on ERK phosphorylation in cells harboring wild-type *BRAF* and, in such cells, it inhibited *TERT* expression more in those harboring *TERT* promoter mutations (Fig. 2a, Supplementary Fig. 2b).

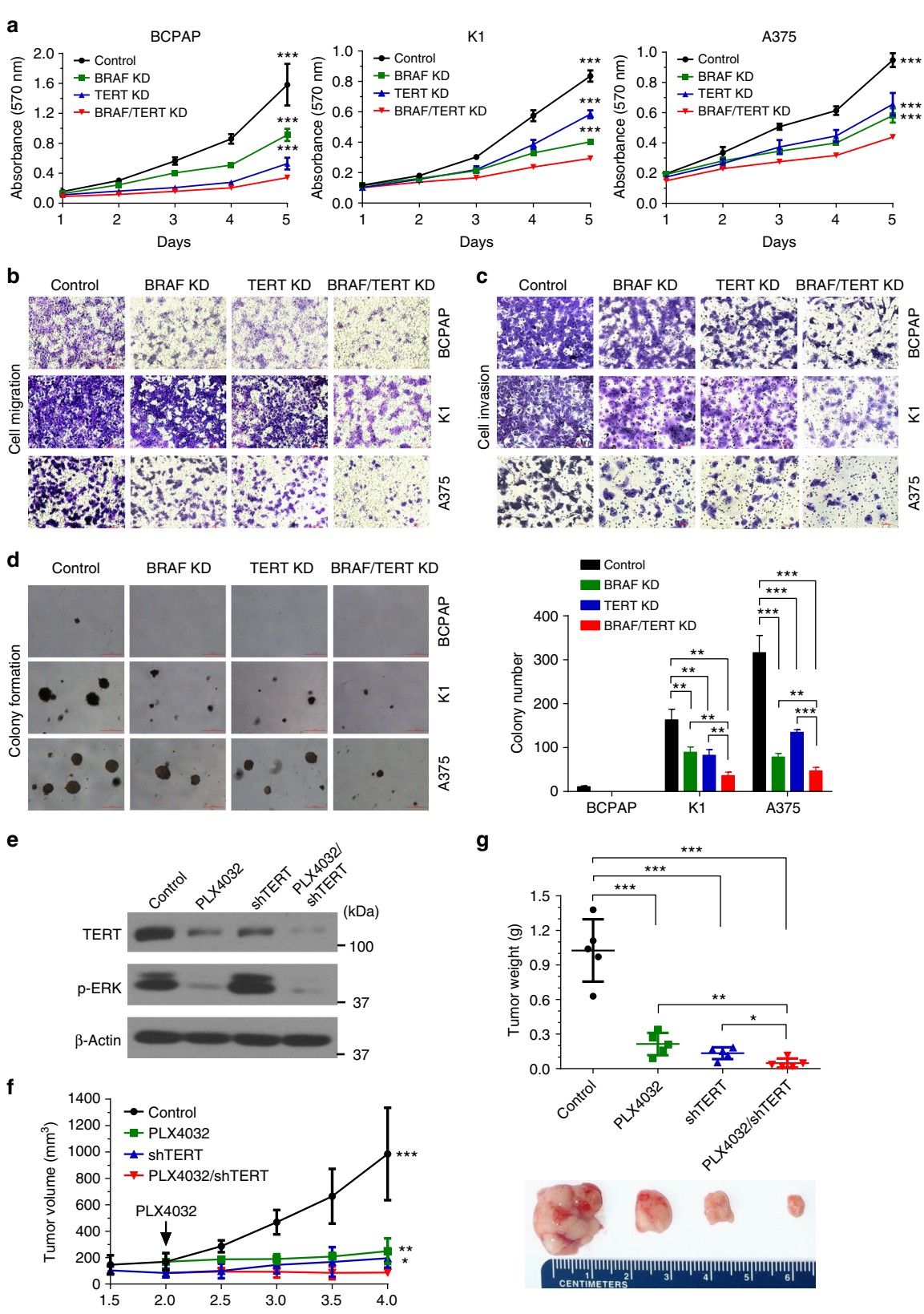

In luciferase reporter assay examining the effect of *BRAF* V600E on *TERT* promoter activities in K1 cells harboring BRAF V600E (Fig. 2b), *TERT* promoter was far more active when harboring the C228T or C250T mutation than the wild-type; treatment with PLX4032 dramatically reduced the activities of the mutated *TERT* promoter and has limited effect on the wild-type *TERT* promoter. To further confirm the role of BRAF V600E in *TERT* expression, we stably knocked down BRAF in BCPAP, K1 and A375 cells, resulting in significantly reduced expression of *TERT* in all these cells (Fig. 2c, d). The somehow less pronounced decrease in TERT protein (Fig. 2c) than the decrease in *TERT* mRNA (Fig. 2d) in the BCPAP cell suggests that the protein translational synthesis system in this cell likely had a good efficiency at low levels of RNA. In contrast, *BRAF* V600E knock-in activated the MAKP pathway (Fig. 2e, f) and increased the activities of *TERT* promoter, especially the mutant types (Fig. 2g). These results again demonstrated a *TERT* promoter mutation-dependent activation of the *TERT* gene by the BRAF V600E/ MAPK pathway.

**Mutation-independent activation of *TERT* by BRAF V600E via MYC.** Since c-MYC was previously shown to activate *TERT* transcription by direct binding to *TERT* promoter[30], we investigated whether MYC played a role in mediating *BRAF* V600E-regulated *TERT* expression. Inhibition of *BRAF* V600E by PLX4032 suppressed *MYC* expression, associated with decreased *TERT* expression in all the cells harboring *BRAF* mutation (Fig. 3a), suggesting that MYC indeed played a role, albeit moderately, in *BRAF* V600E-regulated *TERT* expression. MYC knockdown inhibited *TERT* expression to a similar moderate level as PLX4032 did in RKO and SK-MEL-3 cells which harbored wild-type *TERT* promoter (Fig. 3a). PLX4032 could further inhibit *TERT* expression after siRNA knockdown of MYC in cells harboring *TERT* promoter mutations, but not in cells harboring wild-type *TERT* promoter (Fig. 3a). These data suggest that the *TERT* promoter mutation-dependent activation of the *TERT* promoter by the BRAF V600E/MAPK pathway was a more effective mechanism for the regulation of *TERT* than that by the MYC component and the latter was *TERT* promoter mutation-independent. BRAF V600E knock-in in wild-type *TERT* promoter cells increased MYC expression, which was associated with an increase in TERT expression; the latter increase was abolished by siRNA knockdown of MYC (Fig. 3b). These results provided further evidence that the *TERT* promoter mutation-independent component in the regulation of *TERT* by the BRAF V600E/ MAPK pathway was MYC-mediated. We further showed that *BRAF* V600E knock-in in cells resulted in increased *TERT* promoter activities in a far more robust manner in the promoter harboring the C228T or C250T mutation than the wild-type *TERT* promoter (Fig. 3c). Mutant *TERT* promoter activities were still significantly increased by *BRAF* V600E knock-in in the presence of MYC knockdown, but the wild-type *TERT* promoter activity was not increased by *BRAF* V600E knock-in in the

presence of MYC knockdown (Fig. 3c). These results again suggested that the MYC component in the regulation of *TERT* by the BRAF V600E/MAPK pathway was *TERT* promoter mutation-independent and was minor while the *TERT* promoter mutation-dependent component was dominant.

**BRAF V600E upregulated *GABPB* and GABP binding to mutant *TERT*.** We next investigated how *BRAF* V600E and *TERT* promoter mutations synergistically activated *TERT*. It was recently demonstrated that the ETS transcription factor GABPA selectively bound and activated the mutant *TERT* promoter, but not the wild-type *TERT* promoter, in human cancer cells[31,32]. We hypothesized that the BRAF V600E/MAPK pathway might regulate the GABP transcriptional machinery of *TERT*. To confirm the role of GABP in the regulation of mutant *TERT* in our cell systems, we demonstrated that, similar to GABPA, GABPB also bound to *TERT* promoter in cells harboring *TERT* promoter mutations, but not in cells harboring the wild-type *TERT* (Fig. 4a). Co-immunoprecipitation (Co-IP) assay revealed that GABPA and GABPB formed a complex in the cell (Fig. 4b), consistent with the notion that GABPA complexes with GABPB to form tetramers, creating a fully functional GABP complex that binds DNA and activates gene transcription[33–35]. We next investigated whether BRAF V600E/MAPK pathway regulated the binding of GABP to *TERT* promoter. Chromatin immunoprecipitation (ChIP) assay showed that BRAF knockdown decreased the binding of GABP to the mutant *TERT* promoter when anti-GABPA antibody was used in the assay (Fig. 4c). Interestingly, BRAF knockdown decreased the expression of GABPB, but not GABPA (Fig. 4d), suggesting that the BRAF V600E/MAPK pathway selectively upregulated the *GABPB* gene, resulting in increased production of GABPB, which in turn drove the formation of the GABPA-GABPB transcriptional complex. To further support this concept, luciferase reporter assay showed that BRAF knockdown decreased the promoter activity of *GABPB*, but not that of *GABPA* (Fig. 4e). These results, taken together, demonstrate a mechanism in which by upregulating the expression of *GABPB*, the BRAF V600E/MAPK pathway promotes the formation of the GABPA-GABPB complex and consequent activation of the mutant *TERT* promoter, upregulating *TERT* expression.

**FOS activated the *GABPB* gene by directly binding to its 5′-UTR region.** To decipher how BRAF V600E regulated *GABPB* expression, we took the next step to test our hypothesis that certain target protein molecules of the BRAF/MAPK pathway might function as transcription factors to activate *GABPB*. Our bioinformatics analyses of the regulatory 5′-untranslated region (5′-UTR) of *GABPB* revealed that two classical target molecules downstream of the MAPK pathway, FOS and MYC, were predicted to bind to 5′-UTR of *GABPB* (Fig. 5a). To test this, we constructed a luciferase reporter containing the wild-type 5′-UTR of *GABPB* and two mutant reporters containing disrupted

**Fig.1** Cooperative role of BRAF V600E and TERT in the oncogenic behavior and tumor growth of cancer cells. Specific shRNA against BRAF and siRNA against TERT were used to knock down BRAF and TERT in the indicated cancer cells, respectively, followed by performance of assays of MTT of cell proliferation (**a**), transwell cell migration (**b**), cell invasion (**c**), and colony formation in soft agar (**d**) (with representative images shown in the left panel and the average colony numbers in the right panel). Western blotting analysis of TERT and phosphorylation of ERK (p-ERK) was performed for K1 cells (**e**), from which xenograft tumors were derived to test the role of TERT and BRAF V600E in tumor development and growth (**f**, **g**). Panel **f** shows the time course of tumor growth and panel **g** shows the weights of tumors surgically excised. The "Control" in **a**−**d** represented the combination of scramble shRNA and Control siRNA. The "Control" in **e**−**g** represented the combination of scramble shRNA and DMSO. The horizontal bar in panels **b** and **c** represents 100 μm. The red horizontal bar on the left of panel **d** represents 100 μm; colonies larger than this size were counted and colony numbers are shown on the right of the panel. Little vertical bars in **a**, **d**, **f,** and **g** represent standard deviation (SD). *$P < 0.05$, **$P < 0.01$, ***$P < 0.001$, by two-tailed Student's *t* test. In panel **a**, the *P* values are for the comparison of the indicated condition with "BRAF/TERT knockdown (KD)" (red line). In panel **f**, the *P* values are for the comparison of the indicated condition with "PLX4032/shTERT" (red line)

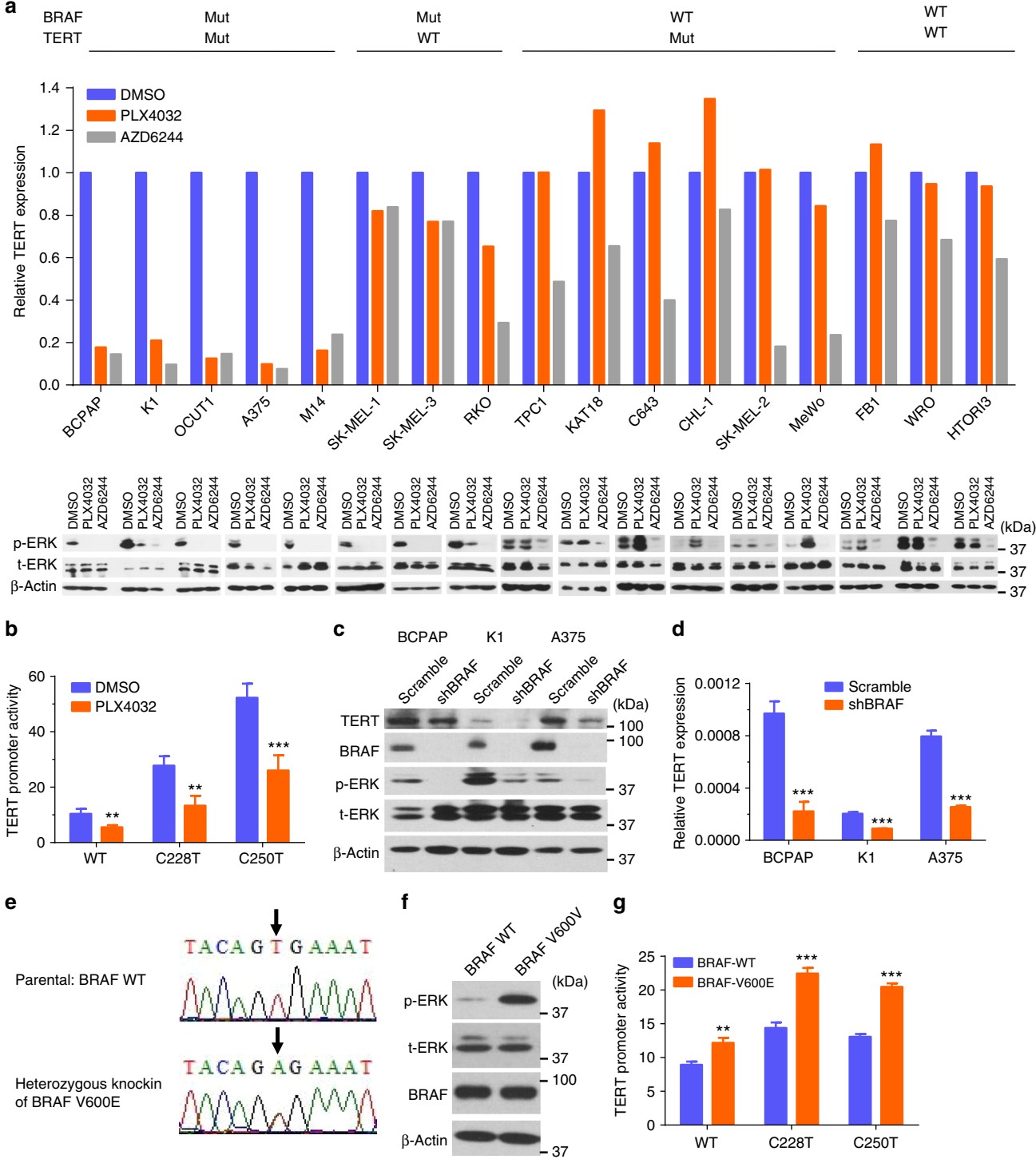

**Fig. 2** *BRAF* V600E/MAPK pathway regulated *TERT* expression. **a** *TERT* expression was analyzed by qRT-PCR in cells treated with 0.5 μM PLX4032 or 0.2 μM AZD6244 for 24 h (upper panel). The corresponding levels of phosphorylated ERK (p-ERK), total ERK (t-ERK), and beta-actin were detected by western blotting (lower panel). The relative *TERT* mRNA expression levels were normalized to the DMSO control group. **b** Luciferase reporting assay of *TERT* promoter activities in K1 cells treated with DMSO or PLX4032 (0.5 μM). **c**, **d** Specific shRNA against *BRAF* were used to knock down BRAF in thyroid cancer cell lines BCPAP and K1 and melanoma cell line A375. Scramble shRNA was used as control. Cells were then subjected to western blotting (**c**) and qRT-PCR (**d**). **e** Sequencing of the *BRAF* exon-15 in the parental and heterozygous *BRAF*-V600E knock-in SW48 cells. *BRAF* V600E was knocked in on one allele of *BRAF* by rAAV technology through homologous recombination and Cre recombinase of the Neo cassette. **f** Western blotting analyses of phosphorylation of ERK (p-ERK), total ERK (t-ERK), BRAF, and beta-actin in the parental and BRAF-V600E knock-in SW48 cells. **g** *TERT* promoter luciferase reporter assay in SW48 cells with/without *BRAF* V600E knock-in. **P < 0.01, ***P < 0.001, by two-tailed Student's *t* test. *P* values are for the comparison of the indicated condition with DMSO (panel **b**), scramble (panel **d**), or BRAF-WT groups (panel **g**). All the values represent the average ± standard deviation (SD) of triplicate samples from a typical experiment. All the experiments were performed three times with similar results

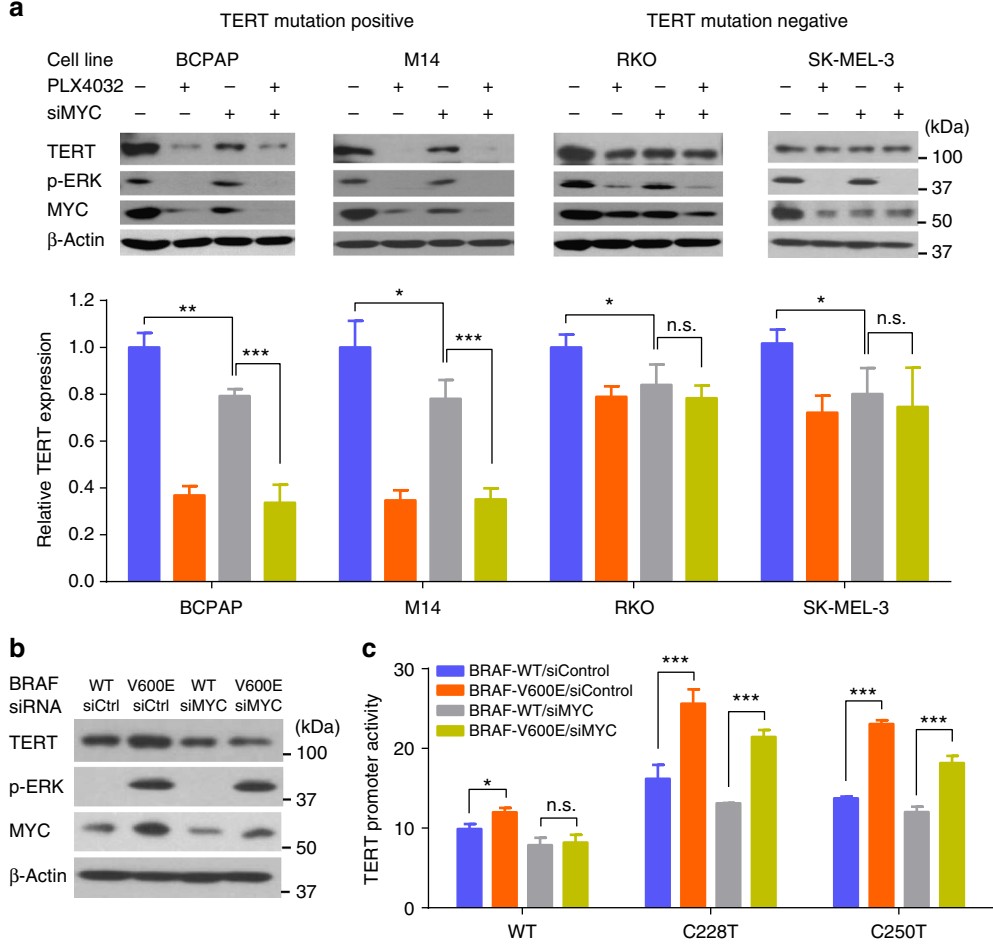

**Fig. 3** MYC-regulated *TERT* expression in a *TERT* promoter mutation-independent manner. **a** The four indicated cancer cell lines harboring *BRAF* V600E mutation were treated with 0.5 μM PLX4032 and/or 10 nM MYC-specific siRNA (siMYC) for 24 h, which were then subjected to western blotting (upper panel) and qRT-PCR (lower panel). The relative *TERT* mRNA expression was normalized to the control group. **b**, **c** Parental and *BRAF* V600E knock-in SW48 cells were treated with control siRNA (siControl) or MYC-specific siRNA (siMYC, 10 nM) for 48 h, followed by western blotting (**b**) and luciferase reporter assays (**c**). *$P < 0.05$, **$P < 0.01$, ***$P < 0.001$, by two-tailed Student's *t* test. n.s. not significant. All the values represent the average ± standard deviation (SD) of triplicate samples from a typical experiment. Similar results were obtained in two additional independent experiments

FOS-binding motif and MYC-binding motif, respectively. If a transcription factor normally binds to the regulatory region and activates a target gene, disruption of the binding site would be expected to lead to decreased activities of the target gene. We found that disruption of the predicted FOS-binding site, but not the MYC-binding site, in the 5′-UTR region of *GABPB* significantly decreased the reporter activities of the *GABPB* gene (Fig. 5b), suggesting that FOS, but not MYC, regulated *GABPB*. We next performed ChIP assay to directly test whether FOS or MYC bound to *GABPB* in the cell and found that FOS, but not MYC, bound to the 5′-UTR of *GABPB* (Fig. 5c). We then used specific shRNA to knock down FOS or MYC and found that knockdown of FOS, but not MYC, could decrease the expression of *GABPB* (Fig. 5d). These results demonstrated that FOS could bind to the 5′-UTR of *GABPB* and activate its expression. We next examined the role of FOS in *TERT* expression. Luciferase reporting assay showed that FOS knockdown specifically inhibited the activities of the mutant *TERT* promoter but not the wild-type *TERT* promoter (Fig. 5e). Correspondingly, FOS knockdown suppressed *TERT* expression in cells harboring the *TERT* promoter mutation, but not in cells harboring the wild-type *TERT* (Fig. 5f). Taken together, these results demonstrated that FOS was a transcription factor of the *GABPB* gene, which could directly bind and activate the promoter of *GABPB*, leading to the

upregulation of *TERT* expression in a *TERT* promoter mutation-dependent manner.

**BRAF V600E pathway promoted phosphorylation and binding of FOS to *GABPB*.** The MAPK pathway was previously shown to stabilize FOS by phosphorylation and stimulate its gene transactivation activities[36,37]. We therefore investigated the role of BRAF V600E in the phosphorylation of FOS. As shown in Fig. 6a, BRAF knockdown decreased the phosphorylation of FOS in cells harboring *BRAF* mutation. Similarly, the BRAF V600E-specific inhibitor PLX4032 significantly decreased the phosphorylation of FOS (Fig. 6b). To test the role of BRAF V600E/MAPK pathway-induced FOS phosphorylation in *GABPB* and mutant *TERT* activation, we overexpressed the wild-type FOS (FOS-wt) and a mutant FOS (FOS-mut) containing alanine replacements on all the phosphorylation sites (Thr-232, Thr-325, Thr-331, and Ser-374) of the target of ERK activation[38]. Overexpression of FOS-wt increased the phosphorylated species of FOS and enhanced the expression of GABPB and TERT (Fig. 6c); it also increased *GABPB* 5′-UTR and mutant *TERT* promoter activities (Fig. 6d, e). In contrast, overexpression of FOS-mut had no effect on *GABPB* and *TERT* activities (Fig. 6c–e). Binding of FOS to *GABPB* was considerably decreased after BRAF knockdown as demonstrated

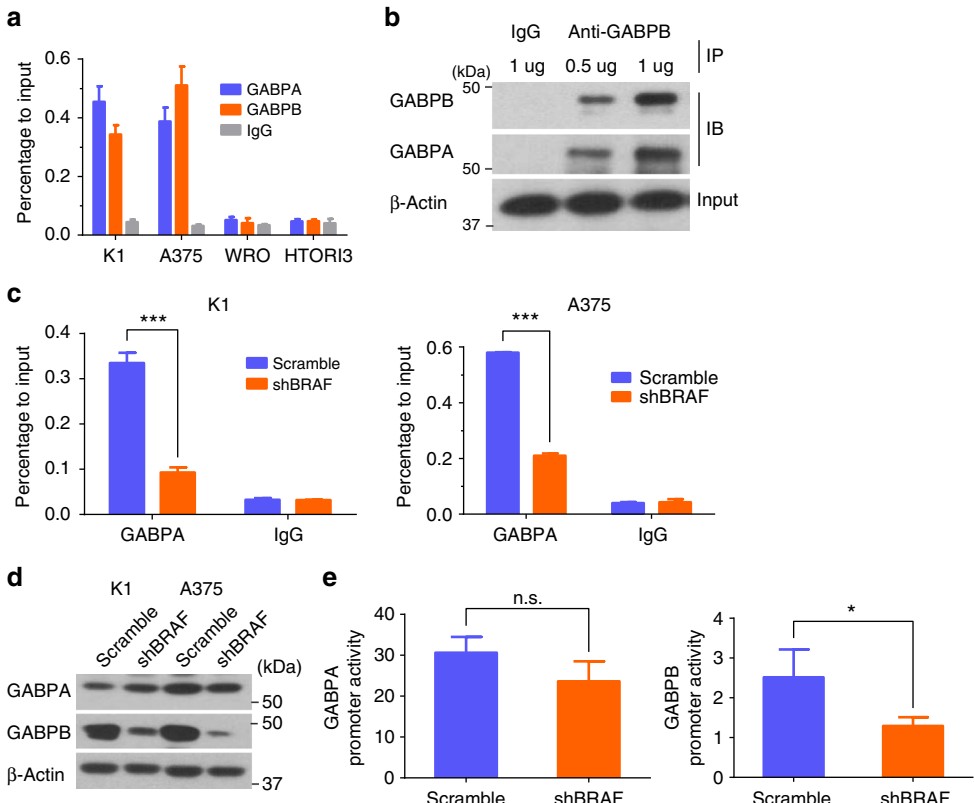

**Fig. 4** *BRAF* V600E promoted GABP binding to mutant *TERT* promoter by upregulating *GABPB*. **a** Chromatin-immunoprecipitation (ChIP) assay for GABPA and GABPB occupancy at *TERT* promoter in K1 and A375 cells with *TERT* promoter mutation and in WRO and HTORI3 cells without promoter mutation. IgG was used as negative control. **b** Co-immunoprecipitation (Co-IP) analysis of the interaction of GABPA with GABPB in K1 cells. **c** ChIP assay for GABPA occupancy at *TERT* promoter in K1 and A375 cells with or without stable BRAF knockdown. **d** Western blotting analyses of GABPA, GABPB, and beta-actin in K1 and A375 cells with/without stable BRAF knockdown. **e** Luciferase reporter assays for *GABPA* and *GABPB* promoters in A375 cells with or without stable BRAF knockdown. Scramble shRNA was used as control. *$P < 0.05$, ***$P < 0.001$, by two-tailed Student's $t$ test. n.s. not significant. All the values represent the average ± standard deviation (SD) of triplicate samples and similar results were obtained in at least two independent experiments

in ChIP assay (Fig. 6f). Conversely, *BRAF* V600E mutation knock-in increased the phosphorylation of FOS (Fig. 6g), enhanced its binding to the 5′-UTR of *GABPB* (Fig. 6h), increased the activities of the *GABPB* promoter (Fig. 6i), and upregulated the expression of the *GABPB* gene (Fig. 6g). Similar results were obtained in WRO cells induced to stably express BRAF V600E (Fig. 6j).

## Discussion

The unique oncogene duet of coexisting *BRAF* V600E and *TERT* promoter mutations is an important recent discovery in human cancer as a robust genetic background for the development of the most aggressive disease in several cancers. For example, it is strongly associated with the most aggressive clinicopathological outcomes of PTC, with hazard ratios for disease recurrence and patient mortality ranging from 30 to 50 compared with patients harboring neither mutation[24,27]; PTC-specific mortality nearly exclusively occurred in patients harboring the oncogene duet of *BRAF* V600E and *TERT* promoter mutations[24,25]. Similar robust synergistic role of this oncogene duet in poor clinicopathological outcomes was seen in other human cancers[22,23]. This genetic duet occurs in about 7–8% PTC[5,24,27] and 20–25% melanoma[9,22], which correspond to the percentages of the cancer cases with the most aggressive diseases and poorest clinical outcomes. These results suggest that this oncogene duet is a superiorly selected genetic event from an evolutionary perspective. As such, it represents a robust genetic mechanism that underpins aggressive

oncogenesis and progression of human cancers and hence ominous clinical outcomes[39]. A fundamental question remains unanswered, however, as to how this oncogene duet cooperates mechanistically, particularly with respect to how BRAF V600E is functionally linked to the mutant *TERT*, in cooperatively driving human cancer aggressiveness.

Our present study brought insights into the mechanism underlying this synergistic oncogenic operation of *BRAF* V600E and *TERT* promoter mutations by demonstrating a robust cooperative role of the two mutations in the expression of *TERT*, through the *BRAF* V600E→MAPK pathway→FOS→GABP→ TERT axis. Using thyroid cancer and melanoma cells as cancer cell models that harbored *BRAF* V600E and *TERT* promoter mutations, we demonstrated that the BRAF V600E/MAPK pathway promoted the formation and binding of transcriptional GABP complex to the mutated *TERT* promoter and its activation. Specifically, we identified *GABPB*, the catalytic unit of the GABP complex, but not the DNA binding unit *GABPA*, as a downstream target gene of the BRAF V600E/MAPK pathway; BRAF V600E-activated MAP kinase pathway strongly upregulated the transcriptional expression of *GABPB*, thus driving the production of GABP complex, which in turn robustly promoted the expression of *TERT*. This represents a major progress in understanding the transcriptional machinery of the mutant *TERT* promoter involving GABP described recently[31,32] by adding a critical regulatory dimension to it.

To gain further molecular insights, we demonstrated that BRAF/MAPK pathway-phosphorylated FOS plays a critical role

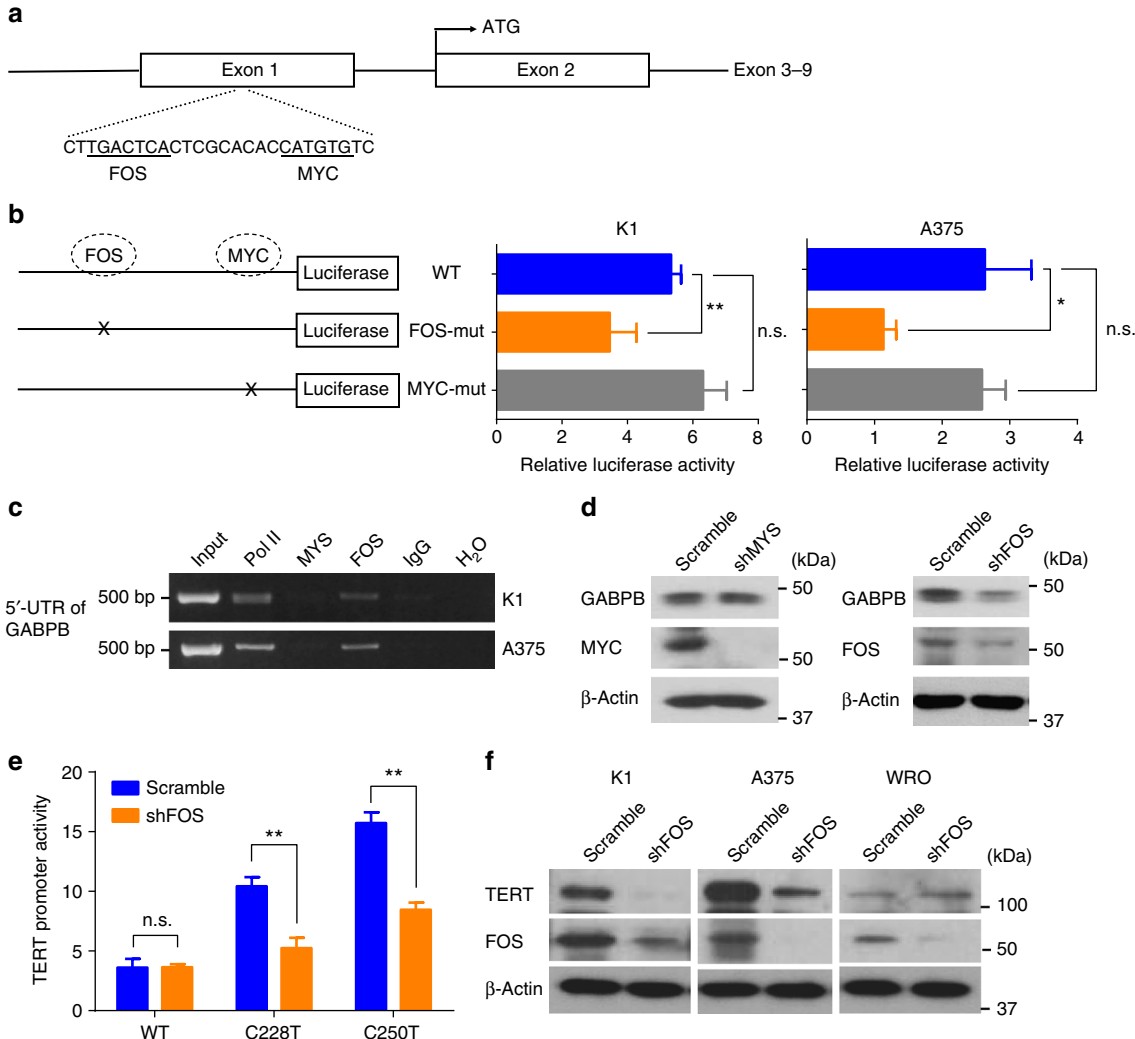

**Fig. 5** FOS bound to 5′-UTR of *GABPB* and upregulated mutant *TERT* expression. **a** Diagrammatic illustration of the putative FOS- and MYC-binding sites in the 5′-UTR of *GABPB* identified by bioinformatics analyses. The predicted palindromic FOS consensus binding site (5′-TGACTCA-3′) and the canonical MYC binding site (5′-CATGTG-3′) located at +222 to +228 and +238 to +243, downstream of the transcriptional start site, respectively. **b** *GABPB* 5′-UTR region-luciferase-reporter assays for the wild-type and artificially-induced mutated putative FOS- and MYC-binding sites. **c** ChIP assay for FOS and MYC occupancy at the 5′-UTR of *GABPB* in K1 and A375 cells. Pol II and IgG were used for positive and negative controls, respectively. **d** Western blotting analyses of GABPB, MYC, FOS, and beta-actin in K1 cells with stable FOS or MYC knockdown. **e** *TERT* promoter-luciferase reporter assays in K1 cells with stable FOS knockdown. **f** Western blotting analyses for TERT, FOS, and beta-actin in K1, A375 and WRO cells with or without stable FOS knockdown. *$P <$ 0.05, **$P <$ 0.01, by two-tailed Student's *t* test. n.s. not significant. All values represent the average ± standard deviation (SD) of triplicate samples, and similar results were obtained in three independent experiments

in this process by acting as a transcriptional factor of the *GABPB* gene. As an initial approach to exploring how BRAF V600E regulated *GABPB*, we performed an in silico analysis, which revealed that both MYC and FOS could bind to the 5′-UTR region of *GABPB*. Our actual experimental test demonstrated that FOS, but not MYC, could bind to the 5′-UTR of *GABPB* and activate its expression. We further demonstrated that *BRAF* V600E/MAPK pathway promoted the phosphorylation of FOS and its binding to the 5′-UTR of *GABPB*, robustly activating *GABPB* and the mutant *TERT*. To directly test if FOS phosphorylation was required for this function of FOS, we engineered FOS to alter the phosphorylation state of FOS and subsequently examined its function in the regulation of *GABP* and *TERT*. Compared with the wild-type FOS, phosphorylation-defective FOS lost the ability to activate *GABPB* and mutant *TERT* in cancer cells, providing direct evidence that phosphorylation of FOS is required for its regulation of *GABPB* and mutant *TERT*.

These findings are consistent with the notion that MAPK/ERK pathway-mediated phosphorylation of FOS is required for its transcriptional activity and transformation efficiency[36–38,40]. The upregulated GABP transcriptional machinery by the BRAF V600E/MAPK pathway mediated by FOS is expected to promote *TERT* promoter mutation-dependent *TERT* expression by facilitating the recruitment and action of classical RNA polymerase. Indeed, this speculation is consistent with a recent study in which RNA polymerase II was found to be recruited to the mutant *TERT* promoter in response to the stimulation by the MAPK pathway[41]. Interestingly, a recent study showed that GABPA bound to the mutant *TERT* promoter mediated long-range chromatin interaction and enrichment of active histone marks as a component of the regulatory machinery for *TERT* transcription[35].

Unlike *GABPB*, we demonstrated that MYC could bind to the promoter of *TERT* and activate it in a *TERT* promoter mutation-

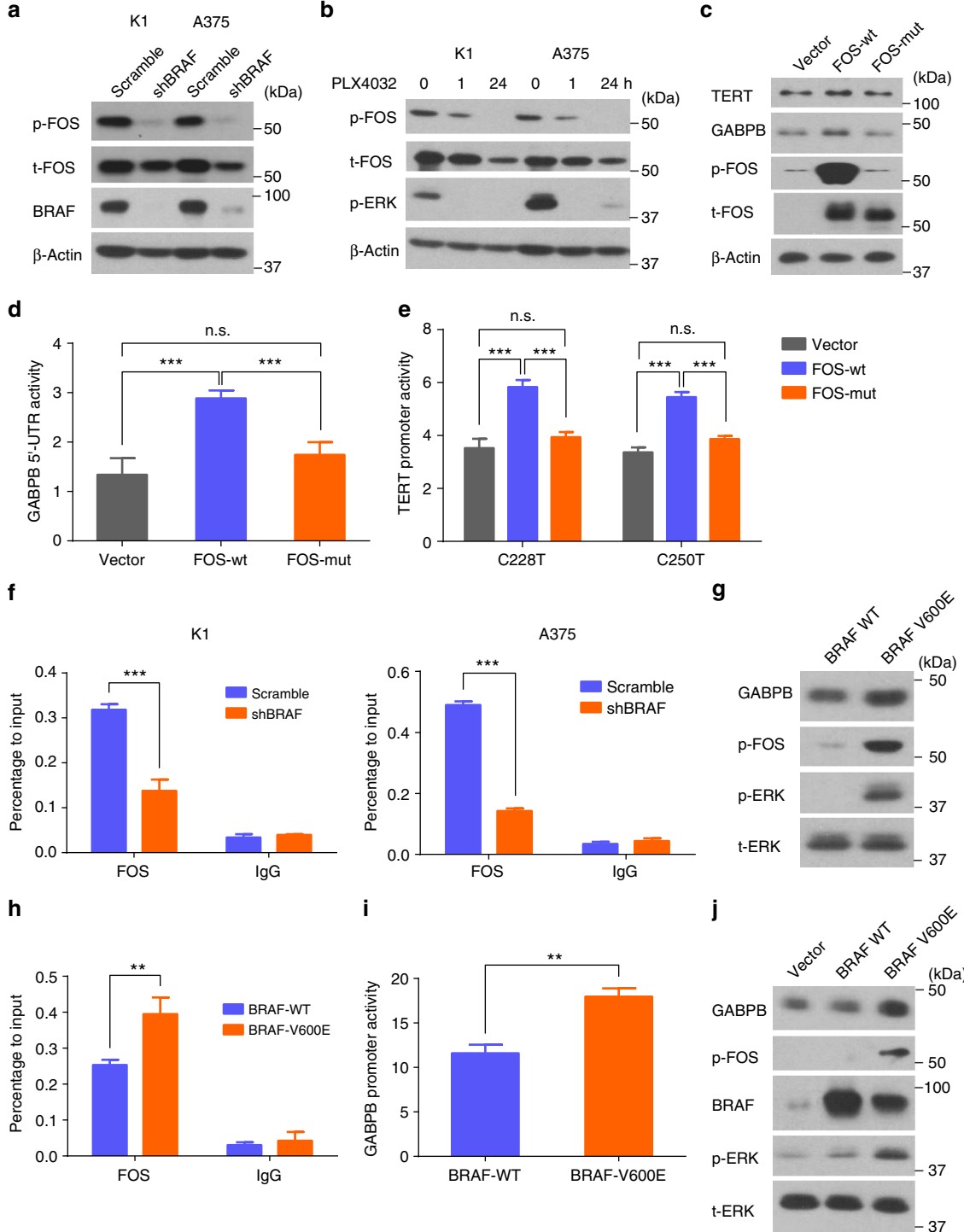

**Fig. 6** *BRAF* V600E promoted FOS binding to *GABPB* by upregulating FOS phosphorylation. **a** Western blotting analyses for phosphorylated-FOS (p-FOS), total FOS (t-FOS), BRAF, and beta-actin in K1 and A375 cells with or without stable BRAF knockdown. **b** Western blotting analyses of p-FOS, t-FOS, p-ERK, and beta-actin in K1 and A375 cells treated with 0.5 μM PLX4032 for 0, 1, and 24 h. **c**, **d**, **e** Effects of FOS phosphorylation on *GABPB* and *TERT* activation. **c** KAT18 cells were serum-starved for 24 h and transiently transfected with FOS wild-type (FOS-wt) or mutant (FOS-mut) bearing none of the potential ERK-targeted phosphorylation sites, followed by western blotting analysis for TERT, GABPB, p-FOS, t-FOS, and beta-actin. **d** KAT18 cells were transfected with FOS-wt or FOS-mut along with GABPB 5'-UTR luciferase reporter and Renilla luciferase (pRL-TK) plasmids in the absence of serum for 24 h, and the luciferase activities were then measured. **e** KAT18 cells were transiently transfected with FOS-wt or FOS-mut, together with *TERT* promoter luciferase reporters harboring the C228T or C250T mutation, and pRL-TK for 24 h, followed by luciferase assays. **f** ChIP assay for FOS binding to the 5'-UTR of *GABPB* in K1 and A375 cells. **g** Western blotting analyses for GABPB, p-FOS, p-ERK, and t-ERK in the parental and *BRAF*-V600E knock-in SW48 cells. **h** ChIP assay for FOS binding to the 5'-UTR of *GABPB* in SW48 cells. **i** *GABPB* 5'-UTR region-luciferase reporter assays in SW48 cells. **j** Wild-type (WT) BRAF and BRAF V600E were stably introduced to express in WRO cells, followed by western blotting analyses of the expression of GABPB, p-FOS, BRAF, p-ERK, and t-ERK after serum starving overnight. **P < 0.01, ***P < 0.001, by two-tailed Student's *t* test. n.s. not significant. All values represent the average ± standard deviation (SD) of triplicate samples and similar results were obtained in two additional independent experiments

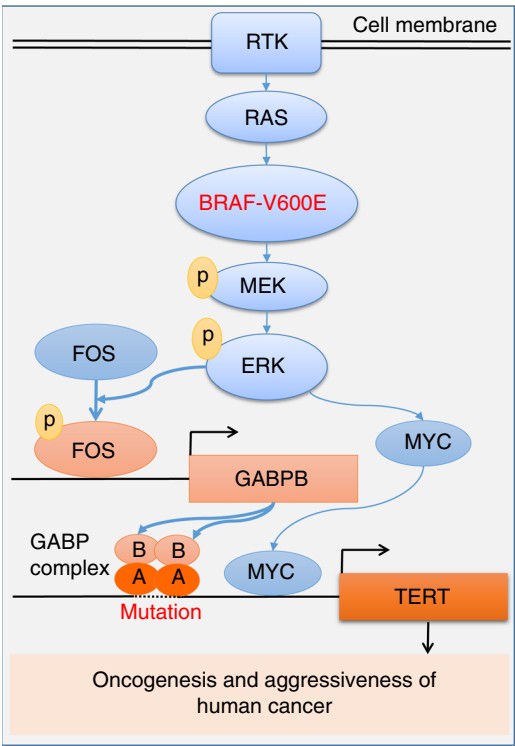

**Fig. 7** Oncogenic cooperation of *BRAF* V600E and *TERT* promoter mutations. This model illustrates a mechanism for the synergistic oncogenicity between *BRAF* V600E and *TERT* promoter mutations in promoting human cancer progression and aggressiveness. This involves promoting *TERT* expression through the *BRAF* V600E → MAPK pathway → FOS → GABPB → GABP complex axis for mutant *TERT* promoter activation; the *TERT* promoter mutation-independent MYC component promoted by the BRAF V600E/MAPK pathway moderately stimulating *TERT* expression is also shown

independent manner. This is consistent with the previous finding that MYC bound to the E-box motif in the *TERT* promoter and upregulated *TERT* expression[30]. The present study demonstrated that the *BRAF* V600E/MAPK pathway could also upregulate *TERT* expression through MYC in a *TERT* promoter mutation-independent manner, but this was less robust than the *TERT* promoter mutation-dependent regulation. This provides a mechanism in which BRAF V600E/MAPK pathway can moderately upregulate *TERT* in cells that do not harbor *TERT* promoter mutations. Thus, the BRAF V600E/MAPK pathway can upregulate the *TERT* gene through both *TERT* promoter mutation-dependent and -independent pathways.

The classical function of TERT is to add telomeres at the end of chromosomes, preventing critical telomere shortening, thus enabling cancer cells acquire replicative immortality[1]. In addition to this function, recent studies have also established TERT as having powerful oncogenic functions in a telomere-independent manner. For example, overexpression of TERT mutant or a naturally occurring alternatively spliced variant of TERT lacking the reverse transcriptase activity stimulated cell proliferation in human and murine cells[42,43]. In fact, enforced TERT expression in transgenic mice promoted the development of spontaneous tumors[44,45]. Even expression of a TERT mutant that retained the catalytic activity but was incapable of maintaining telomere length promoted tumor formation in nude mice[46]. Moreover, TERT activated the RNA-dependent RNA polymerase activity[47], increased cancer cell proliferation by promotes pol III-mediated expression of transfer RNAs[48], and promoted cancer progression

by regulating MYC stability and MYC-dependent oncogenesis[49]. The present study similarly demonstrated a robust oncogenic function of TERT and, importantly, its synergism with BRAF V600E in functioning this way.

In summary and as illustrated in Fig. 7, this study demonstrates for the first time that *BRAF* V600E and *TERT* promoter mutations cooperatively upregulate *TERT* expression via the *BRAF* V600E → MAPK  pathway → FOS → GABP → TERT  signaling/ transcription axis in human cancers. In this process, BRAF V600E/MAP kinase pathway-phosphorylated FOS plays a critical role in oncogenically bridging the *BRAF* V600E and *TERT* promoter mutations by acting as a transcriptional factor of the *GABPB* gene. To a less extent, the BRAF V600E/MAPK pathway also promotes *TERT* expression via MYC in a *TERT* promoter mutation-independent manner. The resulting overexpressed TERT has serious oncogenic consequences. This represents a previously unknown molecular mechanism by which the BRAF V600E/MAPK pathway selectively and robustly regulates the mutant *TERT*, which mechanistically explains the recently observed robust role of the genetic duet of *BRAF* V600E and *TERT* promoter mutations in cooperatively promoting the aggressiveness and poor clinical outcomes of several human cancers. This study holds important cancer biological and clinical implications.

## Methods

**Cell lines**. We used the cell lines originally from the following providers to whom we are very grateful: Thyroid cancer cell line OCUT1 was originally from Dr. Naoyoshi Onoda (Osaka City University Graduate School of Medicine, Osaka, Japan); BCPAP was from Dr. Massimo Santoro (University of Federico II, Naples, Italy); K1 was from Dr. David Wynford-Thomas (University of Wales College of Medicine, Cardiff, UK); TPC1 cell line was from Dr. Alan P. Dackiw (Johns Hopkins University, Baltimore, Maryland); KAT18 was from Dr. Kenneth B. Ain (University of Kentucky Medical Center, Lexington, KY); C643 from Dr. N.E. Heldin (University of Uppsala, Uppsala, Sweden); WRO was from Dr. Guy J. F. Juillard (University of California-Los Angeles School of Medicine, Los Angeles, CA); FB1 was originally from Dr. Fulvio Basolo (Università degli Studi di Pisa, Pisa, Italy). Normal thyroid epithelial cell line-derived HTORI-3 was originally from Dr. N.R. Lemoine (Hammersmith Hospital, London, UK). Melanoma cell lines A375, M14, SK-MEL-1, SK-MEL-2, SK-MEL-3, CHL-1 and MeWo, colon cancer cell line RKO, and human embryonic kidney 293T cells were purchased from American Type Culture Collection (ATCC).

The colon cell line SW48 with heterozygous knock-in of *BRAF* V600E mutation and the parental SW48 cells were purchased from Horizon Discovery (#HD 103-003, Cambridge, UK). The heterozygous knock-in of *BRAF*-activating mutation (V600E) was generated by rAAV technology through homologous recombination and Cre recombinase of the Neo cassette. One allele was knocked in with *BRAF* V600E. It was verified that there was no selection cassette at the engineered locus. BCPAP, K1, OCUT1, TPC1, C643, KAT18, WRO, HTORI-3, M14, and the *BRAF* V600E knock-in and the parental SW48 cells were grown at 37 °C in RPMI-1640 medium with 10% fetal bovine serum (FBS, #F2442; Sigma-Aldrich, St Louis, MO). FB1, A375, CHL-1, and 293T cells were grown at Dulbecco's Modified Eagle Medium (DMEM) medium with 10% FBS. SK-MEL-1, SK-MEL-2, MeWo, and RKO cells were grown in EMEM medium with 10% FBS. The SK-MEL-3 cell was grown in McCoy's 5A medium with 15% FBS. The *TERT* promoter region was amplified by PCR using primers 5′-AGTGGAT TCGCGGGCACAGA-3′ (forward) and 5′-CAGCGCTGCCTGAAACTC-3′ (reverse)[6]; the *BRAF* V600E mutation hot spot region was amplified using primers 5′-TCATAATGCTTGCTCTGATAGGA-3′ (forward) and 5′-GGCCAAAAATT TAATCAGTGGA-3′ (reverse)[50]. The PCR products were subjected to Sanger sequencing for the detection of *BRAF* V600E and *TERT* promoter mutations. The K1 cell line is reported to be contaminated with the GLAG-66 cell line in the International Cell Line Authentication Committee (ICLAC) database. These two cell lines are both human PTC-derived. Our genetic analysis of the K1 cell line used in the present study confirmed typical heterozygous *BRAF* V600E and *TERT* C228T mutations. Cells were authenticated by short tandem repeat analyses and tested for mycoplasma. The K1 cell used here met the purpose of the present study to investigate the role of *BRAF* V600E and *TERT* promoter mutations in human cancer.

**Inhibitors**. The BRAF V600E-specific inhibitor PLX4032 (#S1267) and the MEK1 inhibitor AZD6244 (#S1008) were purchased from Selleck Chemicals (Houston, TX), dissolved in Dimethyl sulfoxide (DMSO) with a stock concentration of 10 mM and stored at −20 °C. PLX4032 and AZD6244 were used to treat cells for 24 h at 0.5

and 0.2 μM, respectively, where indicated in the manuscript. DMSO was used as the vehicle control.

**RNA extraction and quantitative real-time PCR (qRT-PCR)**. Total RNA was extracted from cultured cells using the TRIzol reagent (#15596-018; Ambion, Life Technologies, Carlsbad, CA) and reverse-transcribed to cDNA using the Super-Script III First-Strand Synthesis System (#18080-051; Invitrogen, Life Technologies, Carlsbad, CA). Gene expression was analyzed in triplicate using FastStart Universal SYBR Green Master with ROX (#04913850001; Roche Applied Science, Indiana-polis, IN) on the Applied Biosystems 7900HT Fast Real-Time PCR System. Relative expression of each gene was calculated according to the $2^{-\Delta\Delta Ct}$ method[51]. GAPDH was used as an internal control for normalization. Primers used for qRT-PCR were listed in Supplementary Table 2.

**Western blotting**. Cells were lysed in the RIPA buffer (#sc-24948; Santa Cruz Biotechnology, Santa Cruz, CA) with protease inhibitor cocktail (#P8340; Sigma-Aldrich) and phosphatase inhibitor cocktail (#P0044; Sigma-Aldrich) and western blotting analysis was performed. Briefly, cell lysates were denatured by boiling the sample at 95 °C for 5 min and resolved by sodium dodecyl sulfate polyacrylamide gel electrophoresis (SDS-PAGE). Proteins were transferred to Amersham Hybond-P polyvinylidene difluoride membrane (#10600023; GE Healthcare Life Sciences, Germany) and blocked with 5% non-fat milk in TBS buffer with 0.1% Tween-20 (TBST) at room temperature for 1 h. The membranes were then sliced according to the molecular weights and incubated with primary antibodies at 4 °C overnight, washed with TBST, and incubated with horseradish peroxidase (HRP)-conjugated secondary antibodies at room temperature for 2 h. Signals were detected by SuperSignal™ West Pico PLUS Chemiluminescent Substrate (#34579; Thermo Fisher Scientific). The primary antibodies, including anti-TERT (H-231), anti-BRAF (F-7), anti-ERK (K-23), anti-GABPA (H-180), anti-GABPB (E-7), anti-c-FOS (H-125), anti-phospho-c-FOS (34E4), anti-MYC (9E10), and anti-β-actin (C-4) were purchased from Santa Cruz Biotechnology (Santa Cruz, CA). The anti-phospho-ERK1/2 (Thr202/Tyr204) antibody was purchased from Cell Signaling Technology (Beverly, MA). HRP-linked secondary antibodies, including anti-mouse IgG (#7076S) and anti-rabbit IgG (#7074S), were purchased from Cell Signaling Technology. All uncropped western blotting images are shown in Supplementary Figs. 3–8.

**Transient TERT or MYC knockdown**. TERT siRNA (#sc-36641) and negative control siRNA (#sc-37007) were purchased from Santa Cruz Biotechnology. MYC esiRNA (#EHU021051) and esiRNA targeting EGFP (negative control, #EHUEGFP) were purchased from Sigma-Aldrich. These siRNA were transfected to cells using Lipofectamine RNAiMAX Reagent (Invitrogen) according to the manufactory's protocol. Cells were harvested 2 days after transfection and the knockdown efficiency was determined by western blotting. Transfected cells were subjected to functional analyses.

**Transient FOS and FOS mutant overexpression**. The wild-type FOS cDNA clone was purchased from Origene (#SC116873, OriGene Technologies Inc., Rockville, MD, USA). To construct FOS mutant to prevent phosphorylation, FOS clone containing alanine replacements on Thr-232, Thr-325, Thr-331, and Ser-374 was generated by the QuikChange Lightning Site-Directed Mutagenesis Kit (#210518, Stratagene, La Jolla, CA). The primer sequences used for mutagenesis are listed in Supplementary Table 2. The FOS wild-type and FOS mutant plasmids were transfected into KAT18 cells using Lipofectamine 3000 (Invitrogen) according to the manufactory's protocol.

**Stable knockdown of BRAF, TERT, FOS or MYC in cells**. The short hairpin RNA (shRNA) specifically against BRAF and the scramble control shRNA were cloned into the lentiviral vector pSicoR-PGK-puro (#12084, Addgene, Cambridge, MA)[52]. A pLKO.1-puro based lentiviral vector expressing shRNA against TERT (#TRCN0000240466), FOS (#TRCN0000016007) and MYC (#TRCN0000039642) were purchased from Sigma-Aldrich and the pLKO.1-puro vector with scramble shRNA was purchased from Addgene (plasmid #1864). To generate lentiviral particles, the lentiviral shRNA-expressing vector with the packaging plasmid PSPAX2 and the VSV-G envelope protein-coding plasmid pMD2.G were co-transfected to HEK293T cells using Lipofectamine 3000 (Invitrogen) and the supernatant was harvested 48 h after transfection. To generate cell lines with stable knockdown of BRAF, TERT, FOS or MYC, cancer cells were exposed to the above lentivirus-containing supernatant for 24 h in the presence of 8 μg ml$^{-1}$ polybrene (Millipore, Billerica, MA) and selected by 2 μg ml$^{-1}$ puromycin (Sigma-Aldrich) for 2 weeks. The stable transfection cell pools were confirmed by western blotting analysis of the proteins of interest.

**Introduced BRAF V600E overexpression**. The pBabe-Puro-BRAF-V600E plasmid (Addgene plasmid #15269) was used for introduced overexpression of BRAF V600E in cells that naturally did not harbor BRAF V600E mutation. The BRAF WT was generated from plasmid carrying the V600E mutation (T1799A) using the QuikChange Lightning Site-Directed Mutagenesis Kit (Stratagene, La Jolla, CA)

with primers listed in Supplementary Table 2. Retroviral particles were produced by co-transfecting HEK293T cells with the pBabe-Puro-BRAF-V600E or pBabe-Puro-BRAF-WT plasmid, the packaging plasmid pUMVC, and the envelope plasmid pCMV-VSV-G using Lipofectamine 3000 (Invitrogen). The retroviral supernatant was harvested 48 h after transfection. The WRO cells were infected with a mixture of retrovirus and 8 μg ml$^{-1}$ polybrene (Millipore) and cell pools with stable transfection were selected by 2 μg ml$^{-1}$ puromycin (Sigma-Aldrich).

**Luciferase reporter gene construct and reporter gene assay**. To create the luciferase reporter gene construct containing the TERT promoter, the wide-type core promoter region of TERT, −288 to +61 from the ATG start site, was PCR-amplified from genomic DNA of normal human thyroid cell-derived HTORI-3 cells containing the wild-type TERT promoter. The PCR product was ligated into the pGL3-Basic luciferase vector (Promega, Madison, WI). The resulting plasmid was named p-TERT-WT. This luciferase reporter construct containing the wild-type TERT promoter was induced to contain TERT promoter mutation C228T or C250T by changing the corresponding C allele to T allele using the QuikChange Lightning Site-Directed Mutagenesis Kit (Stratagene). The resulting plasmids were named p-TERT-C228T and p-TERT-C250T, respectively.

A portion of the GABPA promoter (−620 to +268 from the transcription start site) was amplified using genomic DNA and cloned into the pGL3-Basic luciferase vector (Promega). Similarly, the GABPB promoter (−281 to +262 from the transcription start site) was cloned into pGL3-Basic vector. The mutations at the potential FOS and MYC binding sites were generated using the QuikChange Lightning Site-Directed Mutagenesis Kit (Stratagene). The primers used for cloning and mutagenesis are listed in Supplementary Table 2.

For promoter activity assay, cells were seeded in triplicate into a 24-well plate and then transfected with 300 ng pGL3 plasmids containing the TERT, GAPBA, or GABPB promoter, together with 12 ng Renilla luciferase (pRL-TK) plasmid (normalizing control) using Lipofectamine 3000 (Invitrogen). At 24 h after the transfection, cells were lysed and luciferase activities were measured using the Dual-Luciferase Reporter Assay System (Promega). Results were presented as relative luciferase activities, which were obtained by dividing firefly luciferase values with Renilla luciferase values for each set of reading.

**Chromatin immunoprecipitation (ChIP) assay**. The ChIP assay was performed according to the protocol for the fast ChIP method[53]. Briefly, cells were cross-linked with 1% formaldehyde for 10 min, followed by incubation with 125 mM glycine for 5 min at room temperature. Cells were then lysed and sonicated 7 times for 15 s with 45 s rest between pulses at 40% pulse power using a Branson 150D Sonifier Liquid Processor (Branson Ultrasonic Corporation, Danbury, CT). The cross-linked protein/DNA was incubated with anti-FOS, anti-MYC, anti-GABPA, or anti-GABPB antibodies, or non-specific IgG overnight at 4 °C and purified by Protein A/G PLUS-Agarose (sc-2003, Santa Cruz Biotechnology). The precipitated DNA fragments were isolated with Chelex 100 resin (Bio-Rad Laboratories, Her-cules, CA) and subjected to PCR amplification of the GABPB promoter region and the mutation-containing region of the TERT promoter using the primers listed in Supplementary Table 2. The uncropped gel images corresponding to the main figure are provided in Supplementary Fig. 6.

**Co-immunoprecipitation**. Cells were cultured in 100 mm cell culture plates, and lysed in 2.0 ml cold RIPA buffer (sc-24948; Santa Cruz Biotechnology) with pro-tease and phosphatase inhibitors (#P0044; Sigma-Aldrich). Lysates were cen-trifuged at 10,000×g for 10 min at 4 °C and the supernatant was collected. For each immunoprecipitation, a 0.9 ml aliquot of lysate was incubated with 0.5–1.0 μg of the indicated antibody for 1 h at 4 °C and the resulting immuno-complex was pulled down by incubating with 25 μl Protein A/G Plus-Agarose (#sc-2003, Santa Cruz Biotechnology) overnight at 4 °C. The beads were washed four times with 1.0 ml cold lysis buffer, boiled in SDS sample buffer at 95 °C for 5 min, and subjected to western blot analysis using appropriate antibodies. The uncropped images are provided in Supplementary Fig. 5.

**Cell proliferation and colony formation assays**. For cell proliferation assay, cells (800 well$^{-1}$) were seeded on a 96-well plate and MTT assay was carried out daily over a 5-day course to evaluate cell proliferation. At the end of each culture period, 10 μl of the 12 mM MTT (#M6494, Invitrogen) was added to each well. After incubation for 4 h, 100 μl of 10% SDS solution was added, followed by incubation for another 4 h. The absorbance was read at 570 nm. For colony-formation assay, $1 \times 10^3$ K1 cells or $5 \times 10^3$ BCPAP cells were plated in triplicate on a six-well plate with a bottom layer of 0.7% agar and a top layer of 0.35% agar. The total number of colonies ≥100 μm in diameter was counted and representative areas were photo-graphed under a microscope after 3 weeks of culture.

**Cell migration and invasion assays**. Cell migration and invasion assays were performed in triplicates using Transwells in 24-well plates. Transwells with 8-μm pore polycarbonate membrane used for cell migration assay were obtained from Corning (Corning, NY). Transwells coated with Matrigel on the upper surface used for invasion assay were obtained from BD Biosciences (Franklin Lakes, NJ). Cells ($2 \times 10^4$ for migration assay; $5 \times 10^4$ for invasion assay) suspended in 250 μl of

serum-free medium were placed in the upper chamber, while the lower chamber was loaded with 750 µl of cell culture medium with 10% FBS. After 24 h of incubation, the non-invaded cells were removed from the upper surface by a cotton swab. The invaded cells on the lower surface of the membrane were fixed in 100% methanol for 15 min, air-dried, and stained with 0.1% crystal violet. Cells from three microscopic fields were photographed and counted.

**Xenograft tumorigenicity assay**. All animal experiments were approved and performed according to the guidelines of the Institutional Animal Care and Use Committee (IACUC) of Johns Hopkins University. Four-week-old female nude mice (Hsd: Athymic Nude-Foxn1$^{nu}$ mice) were purchased from Harlan Laboratories (Frederick, MD). K1 ($1 \times 10^7$) cells stably expressing scramble shRNA or TERT shRNA were injected subcutaneously into the flanks of nude mice (10 mice per group). At 2 weeks of cell inoculation when the tumors in the control group approached about 150 mm$^3$, each group of animals were divided further into two subgroups (five mice per subgroup) and treated daily with vehicle (5% DMSO, 1% methylcellulose) or 10 mg kg$^{-1}$ PLX4032 by oral gavage. Tumor size was measured twice a week on the skin surface of the animal using a caliper and tumor volume was calculated using the formula (width$^2 \times$ length) $\times 0.5^{54}$. At the end of 4 weeks after cell inoculation, mice were killed and tumors were surgically removed, photographed, and weighted.

**Statistics**. Two-tailed Student's $t$-test was used to determine the significance of difference between two groups in the assays of MTT, cell migration and invasion, colony formation in soft agar, luciferase reporter gene assay, qRT-PCR, and tumor formation in nude mice. For cell migration and invasion assay, colony formation assay, luciferase reporter assay, and qRT-PCR, three independent experiments were carried out, and each was done in triplicate. All the western blotting, Co-IP, and ChIP assays were reproduced at least twice independently with similar results. All $P$ values (Sudent's $t$-test) were two-sided and $P < 0.05$ was considered significant. Analyses were performed using Stata (Stata/SE version 12 for windows; Stata Corp, College Station, TX, USA) and GraphPad Prism (version 6 for Windows; GraphPad Software, Inc., San Diego, CA, USA).

**Data availability**. All the data supporting the findings of this study are available within the article and the Supplementary Information files, or from the authors on reasonable request.

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

## Acknowledgements
Supported by USA NIH grants R01CA113507 and R01CA189224 to M Xing.

## Author contributions
R.L. and M.X. designed the research; R.L., T.Z., G.Z., and M.X. performed the research; R. L. and M.X. analyzed the data; R.L. and M.X. wrote the manuscript, with inputs from all authors; M.X. conceived, supervised and is responsible for the project strategy and management.

## Additional information

**Competing interests:** The authors declare no competing financial interests.

