## [Peer Review File · Nature Communications]

Reviewers' comments:

Reviewer #1 (Remarks to the Author):

The study by Liu et al. aims to investigate the mechanism underscoring the coexistence of BRAF V600E and TERT promoter mutations in promoting the progression of human cancers. The authors demonstrate that BRAF V600E overexpression can phosphorylate and activate FOS, which is a transcriptional activator of GABPB. The transcriptional upregulation of GABPB expression in turn promotes formation of the GABPA-GABPB complex which drives mutant TERT promoter activation. There are already two recent publications last year documenting the direct regulation of BRAF V600E-induced RAS-ERK pathway activation in the transcriptional induction of mutant TERT promoter in human melanomas (Vallarelli et al, *Oncotarget* 7: 53127-53136; Li et al, *PNAS* 113: 14402-14407). Moreover, it is already well established that the GABP heterotetramer complex is a major transcriptional activator of the mutant TERT promoter in human cancers (Bell et al, *Science* 348: 1036-1039). Early studies have also demonstrated that both the GABP alpha and beta subunits are substrates of MAPK/ERK pathway, leading to the regulation of GABP transcriptional activity (Flory et al, *J Virology* 70: 2260-2268). This study lacks novelty and does not go far enough to add new knowledge to the field. Majority of the data shown in Figures 1-4 have already been addressed by existing studies in the field. The authors need to show more mechanistic studies that are novel to justify publication in the journal.

1. 1. It is well known that BRAF V600E signalling is a major driver of melanoma and PTC development, which are two cancers that frequently harbour BRAF mutations. Similarly, upregulation of TERT is a key hallmark of most cancers (~90%). Hence, it is not surprising that knockdown of BRAF and/or TERT will result in reduced cell migration, invasion, proliferation and tumor growth in these cancer cells. Since the authors claim that FOS activation of GABP is a critical regulator of mutant TERT promoter, can the authors indicate if this regulation is dependent on the phosphorylation of FOS and whether a phosphorylation-defective FOS that is engineered in these cells will lead to reduction in tumor growth, migration, etc. and a corresponding reduction in GABPA binding on the mutant TERT promoter?

2. 2. The MAPK/ERK pathway can activate several substrates such as c-Jun, Sp1, ETS1 and GABP which are known to bind and regulate TERT promoter activity. Can the authors address how critical is the regulation of GABP beta by FOS relative to these other factors in the specific activation of mutant TERT promoter?

Minor comments:

1. 1. It is not obvious to the reader that BRAF KD or TERT KD has a significant effect on cell migration and invasion in Figures 1b and 1c.

2. 2. The numbers of colonies shown in Figure 1c are too few to demonstrate the changes in colony formation.

3. 3. SK-Mel-28 cell line has been shown to carry the -57A>C TERT promoter mutation (Vallarelli et al. *Oncotarget* 7: 53127-53136). It is incorrect to show in Figure 2A that this cell line is TERT WT.

Reviewer #2 (Remarks to the Author):

Remarks to the Author:

In this study, Liu et al present evidence for a novel mechanism by which oncogenic BRAF V600E upregulates the expression of TERT specifically in cancer cells harboring mutations in the TERT promoter region. Clinical data have indicated that in certain human cancers the co-existence of the BRAF V600E oncogene and TERT promoter mutations is associated with increased cancer aggressiveness and a poor prognosis. However, the mechanistic link between these two oncogenes is not clarified so far.

By using in vitro and in vivo approaches, the authors have shown that BRAF V600E as an oncogenic driver mutation, activates FOS transcription factor via pERK. Consequently, active (phosphorylated) FOS then upregulates the expression of GABPB, which together in a complex with GABPA acts as a strong transcriptional activator on the TERT promoter, specifically in cancer cells harboring certain point mutations in the TERT promoter region due to preferential binding. This way, the BRAFV600E/ERK axis is mechanistically linked to oncogenic TERT expression, and thus, the co-existence of both oncogenes represents a robust genetic background promoting tumorigenesis and cancer aggressiveness.

The presented data are interesting and of great relevance to the field of intracellular signalling and cancer biology. Therefore, I would recommend this article for being published in Nature Communications after addressing additional aspects mentioned below.

General comments:

(1) The paper is fluently written and the data are nicely discussed. However, given the fact that the co-existence of BRAF V600E and TERT promoter mutations is not just restricted to an aggressive variant of PTC, it would be good to make this a bit more clear in the introduction and in the discussion in order to highlight that these findings are relevant for a better understanding of a broader spectrum of cancer subtypes.

(2) Statistics: in several graphs it is not precisely described for how long the cells were treated, especially in some knockdown experiments - add this information to all Figure legends and/or in the Methods. Please, also include the number of replicates analyzed (technical and biological) and add the statistics, where it is missing in the graphs, otherwise explain in the Figure legend and/or Methods properly.

(3) qPCR experiments: explain how the expression levels were normalized and include this in the Figure legends and/or Methods

(4) Western blot experiments: including total protein levels next to phospho-protein levels (e.g. Fig 1a "total ERK1/2") would strongly support the conclusions.

Specific comments:

(1) Introduction - there is a typo in line 57: "cancer"

(2) Introduction - line 81: are these the same point mutations in all BRAFV600E-driven tumor entities, where the co-existence of TERT promoter mutations is described? Furthermore, the "oncogenic duet" has been described in several types of cancers, but is it in all of them associated

with increased aggressiveness and poor clinical outcome? Please, add here a bit more details to the Introduction.

(3) Introduction: is there a difference between primary tumors and advanced stage/metastasis regarding the co-existence of the so called "oncogenic duet"? - are there any hints indicating that the cooperative effect of both oncogenes is more relevant during metastasis - any data going along this direction should be mentioned in the Introduction or at least in the Discussion?

(4) Figure S1: there are several isoforms of RAF kinases in mammalian cells (ARAF, BRAF, RAF1) and it would be important to visualize ones the specificity of the shRNA against BRAF used in this study.

(5) Fig 1a: is the KD of BRAFV600E stable here? how does the transient TERT KD look like at day 4 or day 5? Is the "control" shown here a control for both, shRNA and siRNA?

(6) Fig 1b + 1c: referring to the statement in line 114 - 116, the decrease in cell migration and invasion for K1 cells and A375 cells comparing the effect of the single KD (BRAF or TERT alone) with the control cannot be clearly concluded from the images shown in both panels. I would suggest to replace them by showing more representative images. In general, the background color of these pictures in both panels varies a lot and reduces the quality. If the data shown here were derived from one complete experiment at a certain time-point, then adjusting the background would improve the quality of both panels.

(7) Fig 1d: instead of showing just a small area of the dishes in the colony formation assay, I would include images showing the full dish with all colonies (if possible). What size does the red bar indicate here?

(8) Fig 1e: does the "control" in the blot represent a technical control for the effect of scrambled shRNA, DMSO or a combination of both?

(9) Fig 1e-1g: please, show DMSO control, scrambled shRNA control and the combination of both separately (at least as a Supplementary information). Because the cells from Fig 1e are derived from the xenograft model, I would go first with the tumor data (Fig 1e "weight", Fig 1f "volume") and then as a last graph the Western blot experiment as Fig 1f.

(10) Fig 2a: PLX4032 was designed to selectively inhibit oncogenic BRAFV600E and it is known that in a BRAF WT genetic background the treatment rather activates ERK. This is also the case in almost all tested BRAF WT/TERT Mut cell lines in this Figure, but it is not described in the Results section. Concluding from BRAF KD and inhibitor experiments presented in this study, active ERK seems to play a crucial role in linking the two oncogenes together, but how strong is this connected to BRAFV600E? E.g. the SK-MEL2 cell line, which was also tested in Fig 2a, harbors a constitutive active NRAS mutation (Q61R) and this cell line is also positive for TERT promoter mutation. Do RAS-mutated cells also show higher TERT expression levels and pERK levels compared to BRAF-WT and RAS-WT cell lines and would a constitutive active RAS mutation together with TERT promoter mutation induce the same phenotype in cancer cells? More hypothetically, is there any scenario possible, where PLX4032 treatment of a BRAFV600E-positive tumor in a patient could potentially promote the malignant transformation/clonal outgrowth of cells in any other tissue/organ harboring somatic TERT promoter mutations in a BRAF WT background? Maybe the authors can speculate about this in the Discussion.

(11) Fig S2: specify what cell line was used here?

(12) Fig 2c: there is a strong discrepancy between TERT protein and RNA expression levels upon BRAF KD in BCPAP cells - are there differences in protein stability - please discuss this better in the manuscript.

(13) Fig 2d: it looks like that TERT RNA expression levels in Fig 2a are normalized to DMSO control, but not in Fig 2d, which brings up the question whether the relatively reduced expression levels in K1 cell line treated with scrambled shRNA are specific for this cell line or rather an effect induced by the scrambled shRNA itself? This could be addressed by including the expression levels in non-treated parental cells.

(14) Fig 2e: the Figure says "BRAFFV600E knockin", but it is not explained, how this knockin was generated - specify this in the Figure legend and explain also in the Methods.

(15) Fig 2g: why has the BRAF WT here such a strong effect on the mutated promoters – it looks to be significant compared to the WT TERT promoter (include statistics comparing the blue bars!) - is this because the cells have also more active ERK meaning that they are above this pERK threshold mentioned earlier?

(16) Fig 3a: the described combinatorial effect of PLX4032 and siMYC on TERT RNA expression in the cells harboring TERT promoter mutations is not visible as mentioned in the Results! How does the TERT expression look like on the protein level? In general, the role of Myc as the authors described in the results and included in the proposed model, is a bit difficult to understand, even more unclear especially if there are no standard deviations included in the bar graph.

(17) Fig 3b-c: I would show first Figure 3c then Figure 3b

(18) Fig 3b: please, show here a lower exposure for beta-Actin and include TERT protein levels

(19) Fig4d: include also BRAF WT vs BRAFFV600E cells to see the difference in GABPB protein levels

(20) Fig 5f: also here a lower exposure for beta-Actin would help to interpret the results better

(21) Fig 6a, 6c and 6f: include BRAF levels in all Western blot experiments to visualize the efficiency of the knockdown.

Responses to Reviewers

To Reviewer #1:

General Comments The study by Liu et al. aims to investigate the mechanism underscoring the coexistence of BRAF V600E and TERT promoter mutations in promoting the progression of human cancers. The authors demonstrate that BRAF V600E overexpression can phosphorylate and activate FOS, which is a transcriptional activator of GABPB. The transcriptional upregulation of GABPB expression in turn promotes formation of the GABPA-GABPB complex which drives mutant TERT promoter activation. There are already two recent publications last year documenting the direct regulation of BRAF V600E-induced RAS-ERK pathway activation in the transcriptional induction of mutant TERT promoter in human melanomas (Vallarelli et al, *Oncotarget* 7: 53127-53136; Li et al, *PNAS* 113: 14402-14407). Moreover, it is already well established that the GABP heterotetramer complex is a major transcriptional activator of the mutant TERT promoter in human cancers (Bell et al, *Science* 348: 1036-1039). Early studies have also demonstrated that both the GABP alpha and beta subunits are substrates of MAPK/ERK pathway, leading to the regulation of GABP transcriptional activity (Flory et al, *J Virology* 70: 2260-2268). This study lacks novelty and does not go far enough to add new knowledge to the field. Majority of the data shown in Figures 1-4 have already been addressed by existing studies in the field. The authors need to show more mechanistic studies that are novel to justify publication in the journal.

Response: There are indeed two papers regarding the regulation of MAPK pathway on TERT expression, but our study was focused on a different and novel mechanism. Li et al. (*PNAS* 2016; 113:14402-14407) focused on the active chromatin state at mutant TERT promoters, and showed that RAS-ERK signaling in BRAF mutant melanomas was involved in the recruitment of RNA polymerase II at mutant TERT promoters. Vallarelli et al (*Oncotarget* 2016;7:53127-53136) focused on ETS1 transcription factor as a mediator between MAPK pathway and TERT expression in melanoma cells. It should be noted that binding of ETS1 to mutant TERT promoter is controversial, at least not universal, in human cancers. First, ETS1/2 was shown to selectively binds to C250T mutant TERT promoter, but not the more common C228T mutant promoter in glioblastoma (Li et al. *Nat Cell Biol* 2015; 17:1327-38). Second, Makowski et al. utilized a proteome-wide approach to identify the interactions between transcription factors in the TERT promoter region, and revealed that GABP was the dominant ETS factor bound to mutant TERT promoter, excluding other ETS factors from binding to the mutant TERT promoter in melanoma (Makowski, et al. *Proteomics* 2016; 16:417-426). Third, two independent groups (Bell et al, *Science* 2015, 348:1036-9; Stern et al, *Genes Dev* 2015, 29:2219-24) screened more than ten ETS factors and identified that GABP, but not others (e.g. ETS1), selectively bound and activated the mutant TERT promoter in multiple cancer types, including melanoma. Taken together, these evidences suggest that GABP transcriptional factor is the most important ETS transcription factor that universally binds to the mutant TERT promoter. Thus, the novel mechanism we have identified in the present study that utilizes FOS to promote GABP and hence TERT is a major mechanism for the upregulation of the mutant TERT, which for the first time links the BRAF V600E/MAPK pathway to mutant TERT through FOS.

Since both BRAF V600E and TERT promoter mutations are highly prevalent and commonly coexist in melanoma and thyroid cancer, we focused on investigating the molecular mechanism of the oncogenic function of this unique genetic duet in human cancer using melanoma and thyroid cancer cells as a model. We focused particularly on whether and how they cooperatively regulated TERT expression. This was to provide a mechanistic explanation for the wide clinical observation of the robust aggressive role of the genetic duet in driving poor clinical outcomes of cancer. Our present study not only confirmed the binding of GABP to mutant TERT promoter in melanoma, but for the first time showed a similar finding in thyroid cancer, providing further evidence for a universal role of GABP in the regulation of mutant TERT promoter. Most importantly, although early studies demonstrated that the MAPK/ERK pathway could regulate phosphorylation of ETS transcription factors, such as GABP (Flory et al, J Virol 70: 2260-2268), whether and how the BRAF V600E/MAPK pathway could regulate the expression of GABP has been unknown. Our present study for the first time found that the protein product of a downstream target gene of BRAF V600E/MAPK, *FOS*, directly bound to the promoter of the *GABPB* gene and upregulated its expression in human cancer, identifying *FOS* as an important novel transcription factor of *GABPB*. This provides a novel molecular mechanism linking the BRAF V600E/MAPK pathway to the specific regulation of mutant TERT, which has never been reported before. Our functional studies from various angles presented in the multiple figures of the manuscript demonstrated that this BRAF V600E/MAPK/*FOS*/*GABP*/mutant *TERT* promoter pathway is indeed an important mechanism in the upregulation of TERT. These studies were logical and necessary to support the conclusions and the mechanistic model proposed in this manuscript. *FOS* plays an important novel role in functionally bridging between BRAF V600E and TERT promoter mutations in human cancers, which is a robust novel finding that moves the field forward an important step, with important novel biological and clinical implications.

Comment 1. 1. It is well known that BRAF V600E signalling is a major driver of melanoma and PTC development, which are two cancers that frequently harbour BRAF mutations. Similarly, upregulation of TERT is a key hallmark of most cancers (~90%). Hence, it is not surprising that knockdown of BRAF and/or TERT will result in reduced cell migration, invasion, proliferation and tumor growth in these cancer cells. Since the authors claim that *FOS* activation of *GABP* is a critical regulator of mutant TERT promoter, can the authors indicate if this regulation is dependent on the phosphorylation of *FOS* and whether a phosphorylation-defective *FOS* that is engineered in these cells will lead to reduction in tumor growth, migration, etc. and a corresponding reduction in *GABPA* binding on the mutant TERT promoter?

Response: To address this point of the reviewer on the role of *FOS* phosphorylation, we have now engineered *FOS* to alter the phosphorylation state of *FOS* and subsequently examined its function in the regulation of *GABP* and *TERT*. Specifically, we observed that compared with the wild-type *FOS*, phosphorylation-defective *FOS* lost the ability to activate *GABPB* and mutant *TERT* in cancer cells. These new results are presented in Figure 6c-e in the revised manuscript and provide further evidence that phosphorylation of *FOS* is required for its regulation of *GABPB* and mutant *TERT*. These findings are also consistent with the established notion that MAPK/ERK pathway phosphorylates *FOS* and the phosphorylation of *FOS* is required for its transcriptional activity and transformation efficiency (Chen et al. PNAS 1993; 90:10952-6. Okazaki et al. EMBO J 1995; 14:5048-59. Monje et al. MCB 2003; 23:7030-43. Monje et al. JBC 2005; 280:35081-4). We added a discussion on this point in the revised manuscript. It

should be noted that the functional studies, including the use of the strategy of BRAF or TERT knockdown as well as cell migration, invasion and proliferation assays presented in multiple figures were all for the ultimate demonstration of the functional importance of the BRAF V600E/MAPK /FOS/GABP/mutant *TERT* promoter signaling system in human cancer. These studies were performed in the context of this novel FOS-mediated mechanism in the specific and robust regulation of mutant TERT by BRAF V600E and hence the synergistic oncogenicity of this genetic duet. The goal of these studies was obviously not to test the function of BRAF V600E and TERT themselves, which have been well established as this reviewer pointed out.

Comment 2. 2. The MAPK/ERK pathway can activate several substrates such as c-Jun, Sp1, ETS1 and GABP which are known to bind and regulate TERT promoter activity. Can the authors address how critical is the regulation of GABP beta by FOS relative to these other factors in the specific activation of mutant TERT promoter?

Response: Although the MAPK/ERK pathway can active several ETS1 and many other transcriptional factors, as discussed above GABP is the dominant and universal transcription factor for *TERT* and is *TERT* promoter mutation-dependent. We therefore focused particularly on GABP in the context of the role of FOS and the dependence on *TERT* promoter mutation. For example, we demonstrated that FOS knockdown significantly decreased the GABPB expression (Fig 5d), activity of mutant TERT promoter (Fig 5e), and TERT expression in cells harboring *TERT* promoter mutations (Fig 5f).

Minor comments:

Comment 1. 1. It is not obvious to the reader that BRAF KD or TERT KD has a significant effect on cell migration and invasion in Figures 1b and 1c.

Response: If specifically comparing with the corresponding controls, it can actually be seen that BRAF KD or TERT KD significantly decreased cell migration and invasion (Fig 1b and 1c). Following the suggestion of Reviewer 2 (see below), we now present representative images with adjusted backgrounds for these figures in the revised manuscripts, which visually more clearly show the effects of KD.

Comment 2. 2. The numbers of colonies shown in Figure 1c are too few to demonstrate the changes in colony formation.

Response: In fact, hundreds of colonies occurred in the control groups for K1 and A375 cells; BRAF and/or TERT knockdown significantly decreased colony formation and the colony numbers were correspondingly less than the control groups. Although only a few colonies formed with BCPAP cells, knockdown of either BRAF or TERT completely abolished the colony formation, showing the strong oncogenic function of BRAF and TERT. The accompanying bar graph even more clearly showed the changes in colony formation.

Comment 3. 3. SK-Mel-28 cell line has been shown to carry the -57A>C TERT promoter mutation (Vallarelli et al. *Oncotarget* 7: 53127-53136). It is incorrect to show in Figure 2A that this cell line is TERT WT.

Response: We thank the reviewer for reminding us that Vallarelli et al found the -57A>C TERT promoter mutation in the SK-MEL-28 cell. We also note that Bell et al reported that this cell line was TERT wild-type (Bell et al. *Science* 2015; 348:1036-9). To avoid this controversy,

we have now replaced this cell line with SK-MEL-3 cell which does not harbor any *TERT* promoter mutation and repeated the indicated experiments. The revised figure using the SK-MEL-3 cell is included in the revised manuscript. This point of the reviewer alerted us to potentially a similar issue with other cell lines. We therefore have now examined all the cell lines in the present study and found no -57A>C mutation in any of them.

Responses to Reviewer #2

Overall comments of the Reviewer:

In this study, Liu et al present evidence for a novel mechanism by which oncogenic BRAF V600E upregulates the expression of TERT specifically in cancer cells harboring mutations in the TERT promoter region. Clinical data have indicated that in certain human cancers the co-existence of the BRAF V600E oncogene and TERT promoter mutations is associated with increased cancer aggressiveness and a poor prognosis. However, the mechanistic link between these two oncogenes is not clarified so far.

By using in vitro and in vivo approaches, the authors have shown that BRAF V600E as an oncogenic driver mutation, activates FOS transcription factor via pERK. Consequently, active (phosphorylated) FOS then upregulates the expression of GABPB, which together in a complex with GABPA acts as a strong transcriptional activator on the TERT promoter, specifically in cancer cells harboring certain point mutations in the TERT promoter region due to preferential binding. This way, the BRAFV600E/ERK axis is mechanistically linked to oncogenic TERT expression, and thus, the co-existence of of both oncogenes represents a robust genetic background promoting tumorigenesis and cancer aggressiveness.

The presented data are interesting and of great relevance to the field of intracellular signalling and cancer biology. Therefore, I would recommend this article for being published in Nature Communications after addressing additional aspects mentioned below.

Response: We thank this reviewer for the accurate and concise summary of the current state of the filed in relation to our study and our findings. We also thank this reviewer for the kind comments about the mechanistic novelty and biological and clinical relevance of our findings.

General comments:

(1) The paper is fluently written and the data are nicely discussed. However, given the fact that the co-existence of BRAF V600E and TERT promoter mutations is not just restricted to an aggressive variant of PTC, it would be good to make this a bit more clear in the introduction and in the discussion in order to highlight that these findings are relevant for a better understanding of a broader spectrum of cancer subtypes.

Response: We thank the reviewer for this very useful advice, following which we have now added more details on the association of the genetic duet of coexisting BRAF V600E and TERT promoter mutations with poor clinical outcomes in cancers other than PTC, especially in melanoma.

(2) Statistics: in several graphs it is not precisely described for how long the cells were treated, especially in some knockdown experiments - add this information to all Figure legends and/or in the Methods. Please, also include the number of replicates analyzed (technical and biological) and add the statistics, where it is missing in the graphs, otherwise explain in the Figure legend and/or Methods properly.

Response: The information has now been added in the corresponding figure legends and in the statistics section of Methods.

(3) qPCR experiments: explain how the expression levels were normalized and include this in the Figure legends and/or Methods

Response: This information is now added in the corresponding figure legends and Methods.

(4) Western blot experiments: including total protein levels next to phospho-protein levels (e.g. Fig 1a "total ERK1/2") would strongly support the conclusions.

Response: The total ERK expression is now added in Figures 2a, 2c, 2f, 6g, and 6j. The total FOS expression is shown in Figures 6a, 6b and 6c.

Specific comments:

(1) Introduction - there is a typo in line 57: "cancer"

Response: The typo error has been corrected.

(2) Introduction - line 81: are these the same point mutations in all BRAFV600E-driven tumor entities, where the co-existence of TERT promoter mutations is described? Furthermore, the "oncogenic duet" has been described in several types of cancers, but is it in all of them associated with increased aggressiveness and poor clinical outcome? Please, add here a bit more details to the Introduction.

Response: *BRAF* V600E is the most common type of point mutation of *BRAF* in human cancer and virtually all studies focused on the co-existence of *BRAF* V600E and *TERT* promoter mutations. The association of this oncogenic duet with tumor aggressiveness and poor clinical outcomes has been well established in thyroid cancer and melanoma. We therefore focused on these two cancers in the Introduction section.

(3) Introduction: is there a difference between primary tumors and advanced stage/metastasis regarding the co-existence of the so called "oncogenic duet"? - are there any hints indicating that the cooperative effect of both oncogenes is more relevant during metastasis - any data going along this direction should be mentioned in the Introduction or at least in the Discussion?

Response: A number of studies have shown a strong association between the oncogenic duet and advanced tumor stages and lymph node and distant metastases in thyroid cancer. We have now discussed this in more details in the Introduction.

(4) Figure S1: there are several isoforms of RAF kinases in mammalian cells (ARAF, BRAF, RAF1) and it would be important to visualize ones the specificity of the shRNA against BRAF used in this study.

Response: The shRNA for BRAF stable knockdown used in this study was specific for BRAF, which has no homology sequence to other genes (including other isoforms of RAF kinases). In fact, this shRNA has been widely used for specific BRAF knockdown in other studies. We cited one of these studies as a reference in the Method.

(5) Fig 1a: is the KD of BRAFV600E stable here? how does the transient TERT KD look like at day 4 or day 5? Is the "control" shown here a control for both, shRNA and siRNA?

Response: Yes, the KD of BRAF was stable. The TERT expression was significantly suppressed at day 2 after TERT siRNA transfection as shown in Figure S1. The aim of this section of studies was to show the combinational effect of BRAF V600E and TERT on cell behaviors. To this end, the results in Figure 1a-d are sufficient to show the combinational oncogenic functions of the two oncogenes. Therefore, we did not test the TERT expression at day 4 or day 5. The “control” in Fig 1a-d represented combined control treatments for both shRNA and siRNA.

(6) Fig 1b + 1c: referring to the statement in line 114 - 116, the decrease in cell migration and invasion for K1 cells and A375 cells comparing the effect of the single KD (BRAF or TERT alone) with the control cannot be clearly concluded from the images shown in both panels. I would suggest to replace them by showing more representative images. In general, the background color of these pictures in both panels varies a lot and reduces the quality. If the data shown here were derived from one complete experiment at a certain time-point, then adjusting the background would improve the quality of both panels.

Response: Following the reviewer’s suggestion, we presented more representative images in the revised manuscript. Since all the data were derived from one complete experiment, we adjusted the background of the images to improve the quality. The difference between KD and control is more evident visually.

(7) Fig 1d: instead of showing just a small area of the dishes in the colony formation assay, I would include images showing the full dish with all colonies (if possible). What size does the red bar indicate here?

Response: These images were to show that the sizes of colonies were different among the four groups. To this end, a magnified small area of the dishes (left panel) showed most clearly the sizes of colonies. Image of a whole dish, which would have to be in low magnification, showed small colonies with indistinguishable sizes. To further quantitate the colonies, we counted the number of all the colonies in each dish, which are presented in the right panel. The red bar represents 100 μ m and colonies larger than this size were counted.

(8) Fig 1e: does the "control" in the blot represent a technical control for the effect of scrambled shRNA, DMSO or a combination of both?

Response: The “*in vivo* control” in Fig 1e-g represented the combination of scramble shRNA and DMSO. We added this information in the figure legend.

(9) Fig 1e-1g: please, show DMSO control, scrambled shRNA control and the combination of both separately (at least as a Supplementary information). Because the cells from Fig 1e are derived from the xenograft model, I would go first with the tumor data (Fig 1e "weight", Fig 1f "volume") and then as a last graph the Western blot experiment as Fig 1f.

Response: These experiments represent *in vivo* studies on xenograft tumors growing from cells without knockdown (control) or knockdown of the indicated molecule(s). No DMSO or shRNA was administered in the animals. Cells in Fig 1e were not derived from xenograft tumors; in fact, they were cells used to inoculate nude mice to produce xenograft tumors. The order of the panels is therefore logical.

(10) Fig 2a: PLX4032 was designed to selectively inhibit oncogenic BRAFV600E and it is known that in a BRAF WT genetic background the treatment rather activates ERK. This is also

the case in almost all tested BRAF WT/TERT Mut cell lines in this Figure, but it is not described in the Results section. Concluding from BRAF KD and inhibitor experiments presented in this study, active ERK seems to play a crucial role in linking the two oncogenes together, but how strong is this connected to BRAFV600E? E.g. the SK-MEL2 cell line, which was also tested in Fig 2a, harbors a constitutive active NRAS mutation (Q61R) and this cell line is also positive for TERT promoter mutation. Do RAS-mutated cells also show higher TERT expression levels and pERK levels compared to RAF-WT and RAS-WT cell lines and would a constitutive active RAS mutation together with TERT promoter mutation induce the same phenotype in cancer cells? More hypothetically, is there any scenario possible, where PLX4032 treatment of a BRAFV600E-positive tumor in a patient could potentially promote the malignant transformation/clonal outgrowth of cells in any other tissue/organ harboring somatic TERT promoter mutations in a BRAF WT background? Maybe the authors can speculate about this in the Discussion.

Response: As the reviewer pointed out and as can be expected, PLX4032 treatment indeed activated ERK accompanied by increased TERT expression in several BRAF WT cells in our study. We have now described this result in the Result section in the revised manuscript. Our results indeed suggest that ERK plays a crucial role in linking BRAF V600E and TERT promoter mutations.

The RAS mutation (especially NRAS) mutation has been shown to be associated with TERT promoter mutations. Our recent studies showed that coexistence of RAS and TERT mutations was associated with aggressiveness of human tumor and poor clinical outcomes (Shen, et al. *Endocr Relat Cancer*. 2017, 24(1):41-52). Since mutant RAS could activate both MAPK and PI3K pathways by phosphorylating ERK and AKT, respectively, it is likely that RAS-mutant cells had a higher TERT expression than the RAS wild-types ones. Further studies are needed to investigate this possibility.

That PLX4032 could potentially activate mutant TERT in RAF WT cells and hence promoter transformation of such cells is an interesting hypothetical speculation.

The above are not the focus of the present study. To avoid distraction, we choose not to extend the discussion on them in the manuscript; the focus of the present study is the novel mechanism related to the BRAF V600E/MAPK/FOS/GABP/TERT axis. We respectfully hope that the reviewer agrees with us.

(11) Fig S2: specify what cell line was used here?

Response: The cells harboring both BRAF mutation and TERT mutation includes BCPAP, K1, OCUT1, A375, and M14; the cells harboring BRAF mutation but not TERT mutation includes SK-MEL-1, SK-MEL-3, and RKO; the cells harboring TERT mutation but not BRAF mutation includes TPC1, KAT18, C643, CHL-1, SK-MEL-2, and Mewo; the cells harboring no BRAF or TERT mutation includes FB1, WRO, and HTORI3. We have now added this information in the Figure legend.

(12) Fig 2c: there is a strong discrepancy between TERT protein and RNA expression levels upon BRAF KD in BCPAP cells - are there differences in protein stability - please discuss this better in the manuscript.

Response: Both TERT protein and mRNA levels were significantly decreased after BRAF KD in all the three cell lines. As TERT expression was naturally low in K1 cells, the film exposure time of the whole membrane had to be longer than ideal, artificially lessening the visual

difference of the protein signals. We also had films with shorter exposure time in which the difference of TERT protein looked more prominent in BCPAP cells but the naturally weak signal in K1 cells became invisible. With a more appropriate exposure, the film shows a more clear decrease in TERT protein after BRAF KD. The somehow less pronounced decrease in TERT protein (Fig 2c) than the decrease in TERT RNA (Fig 2d) suggests that the protein translational production system in the BCPAP cell was likely sufficiently efficient at low levels of RNA. This is now stated in the revised manuscript.

(13) Fig 2d: it looks like that TERT RNA expression levels in Fig 2a are normalized to DMSO control, but not in Fig 2d, which brings up the question whether the relatively reduced expression levels in K1 cell line treated with scrambled shRNA are specific for this cell line or rather an effect induced by the scrambled shRNA itself? This could be addressed by including the expression levels in non-treated parental cells.

Response: We tested the level of TERT mRNA expression in non-treated cells and found it to be very similar to that seen here in the scramble group. To keep the figure style consistent with others (figure 2b, 2c, 2g), TERT mRNA levels in the non-treated parental cells were not shown here.

(14) Fig 2e: the Figure says "BRAFV600E knockin", but it is not explained, how this knockin was generated - specify this in the Figure legend and explain also in the Methods.

Response: SW48 with heterozygous knock-in of BRAF V600E mutation and the paired parental SW48 cells were directly purchased from Horizon Discovery ltd. It was generated by rAAV technology through homologous recombination and Cre recombinase of the Neo cassette. We added this information in figure legend and Methods.

(15) Fig 2g: why has the BRAF WT here such a strong effect on the mutated promoters – it looks to be significant compared to the WT TERT promoter (include statistics comparing the blue bars!) - is this because the cells have also more active ERK meaning that they are above this pERK threshold mentioned earlier?

Response: We wish to respectfully remind that what the reviewer observed is actually a well-known phenomenon—the basal activity of the mutant TERT promoter is significantly higher than that of the wild-type TERT promoter, which is seen here and consistent with the results in Fig 2b and in many previous publications as well. This is consistent with the function of the TERT promoter mutation— increase in the TERT promoter activity. So, it does not represent “a strong effect of BRAF WT”.

(16) Fig 3a: the described combinatorial effect of PLX4032 and siMYC on TERT RNA expression in the cells harboring TERT promoter mutations is not visible as mentioned in the Results! How does the TERT expression look like on the protein level? In general, the role of Myc as the authors described in the results and included in the proposed model, is a bit difficult to understand, even more unclear especially if there are no standard deviations included in the bar graph.

Response: Indeed MYC represented a minor but non-negligible pathway compared with the BRAF V600E/MAPK/FOS/GABP pathway as we proposed in the model. This is consistent with the overall data in the present study. We have now included the TERT protein expression in Fig 3a and the standard deviations to more clearly show the modest but visible effect of MYC.

Statistics are also added. The effect of MYC is now clearly presented. We also added more description on this result in the revised manuscript.

(17) Fig 3b-c: I would show first Figure 3c then Figure 3b

Response: We presented Figure 3b first to show that both the BRAF V600E knock-in and MYC transient knockdown were successful, which was basis for the subsequent studies presented in the figure. Therefore, the order of the figure panels is logical.

(18) Fig 3b: please, show here a lower exposure for beta-Actin and include TERT protein levels

Response: A revised version of the figure with beta-actin, and TERT protein expression added is now presented in for figure 3b.

(19) Fig4d: include also BRAF WT vs BRAFV600E cells to see the difference in GABPB protein levels

Response: Studies in Fig 4d were performed to see whether GABPA and GABPB were differentially regulated by the BRAF V600E using knockdown approach. Therefore BRAF V600E-harboring cells were used and the goal of the study was achieved. It is not necessary and adds no help to knock down BRAF in BRAF WT cells. We respectfully hope that the reviewer agrees with us.

(20) Fig 5f: also here a lower exposure for beta-Actin would help to interpret the results better

Response: The figures with protein expressions have now been revised by including lower exposures.

(21) Fig 6a, 6c and 6f: include BRAF levels in all Western blot experiments to visualize the efficiency of the knockdown.

Response: Fig 6b-e are now revised with a new version of figure 6. The original 6c and 6f are changed to 6g and 6j, respectively. The BRAF protein levels have been added to Fig 6a and 6j. The BRAF protein level in the BRAF V600E knock-in cell (6g) is shown in Fig 2f.

Again, we thank the two reviewers for the critical review of the manuscript and the constructive critiques, which we found to be extremely helpful in guiding our revision and improving of the manuscript.

REVIEWERS' COMMENTS:

Reviewer #1 (Remarks to the Author):

The revised paper by Liu et al has addressed a number of my concerns. Since the discovery of TERT promoter mutations by a few of our colleagues, it has become clear that the mechanism of activation of this promoter is also an important facet in itself. Given that there are only a few papers on this rapidly evolving field, I would like to make some suggestions regarding the nomenclature and literature such that we do not have to correct the literature later. This exercise will also provide the readers a more balanced and educated view going forward. Here are my suggestions.

1) I would like the authors to refer the mutations as per their position (-146C>T and -124 C>T as referred by Horn et al., Science 2013 (339:p959-961). The nomenclature C228T and C250 T is incorrect from genomics perspective.

2) It was also shown that mutant TERT promoters also bind NFkB proteins (although the authors of this study also incorrectly refer to this mutation as C250T). This must be mentioned in paragraph 1 of introduction.

3) Line numbers 35 and 288 look strikingly similar and must be changed.

4) Line number 359 mentions telomere independent roles of TERT and in this context it must be mentioned that Tert can regulate RNA dependent RNA polymerase activity Maida et al., Nature 2009 and tRNA synthesis (Khattar et al., JCI 2016). It is also interesting that genetic evidence shows that TERT regulates myc dependent oncogeneis independent of its telomerase activity Koh et al JCI 2015 and this in my view is the first in vivo evidence and genetic evidence of a non-canonical role of TERT.

5) A major implication of this study could be that therapy for cancers with both BRAf and TERT mutations could depend on each other. Indeed modeling genomic instability and selection pressure in murine melanoma by Kwong et al Cell Reports 2017 will be important to quote.

6) Finally, in the model (Figure 7), the authors show the GABPA dimers bind on the mutant Tert promoter. But a Cancer Discovery paper by AKincilar et al 2016 showed that TERT promoter can also be bound by tetramer of GABPA from a long distance. This is very much in line with data from Steve McKnight and others that GABPA proteins tetramerise for activation. Indeed, regulation of TERT by long distance promoter looping has been demonstrated by Kim et al., Plos Biology 2016. These papers must be suitably incorporated for correctness.

Reviewer #2 (Remarks to the Author):

I would like to thank the Authors at this point for taking up most of the Reviewer's suggestions. The present study is novel enough and the revised version of the manuscript includes enough additional details strengthening the conclusions of this study. All major and minor comments were addressed convincingly or at least well explained, especially the statistics section has been improved. In addition, all experiments are well controlled and state-of the art methods were used throughout the study. Furthermore, the data presented by Liu et al deal with a clinically extremely relevant topic, the data are solid, and therefore, I am suggesting the revised version of this manuscript for publication in Nature Communications.

Responses to Reviewers

To Reviewer #1

General Comment: The revised paper by Liu et al has addressed a number of my concerns. Since the discovery of TERT promoter mutations by a few of our colleagues, it has become clear that the mechanism of activation of this promoter is also an important facet in itself. Given that there are only a few papers on this rapidly evolving field, I would like to make some suggestions regarding the nomenclature and literature such that we do not have to correct the literature later. This exercise will also provide the readers a more balanced and educated view going forward. Here are my suggestions.

Response: We thank this reviewer for the careful review of our manuscript and the helpful comments.

Comment 1) I would like the authors to refer the mutations as per their position (-146C>T and -124 C>T as referred by Horn et al., *Science* 2013 (339:p959-961). The nomenclature C228T and C250 T is incorrect from genomics perspective.

Response: We appreciate the suggestion of the reviewer. In deed, some studies refer the two mutations as -146C>T and -124 C>T based on their positions with respect to the translation start site of *TERT*. The nomenclature of C228T and C250T, taking the last three digits of the nucleotide number where the reference nucleotide is changed on chr 5, in fact came from one of the two original studies discovering *TERT* promoter mutations (Huang, et al., *Science*. 2013;339(6122):957-9). They are widely accepted and used in the scientific literature for its simplicity [for example, Killela, et al., *PNAS*. 2013;110(15):6021-6. Bell, et al., *Science*. 2015;348(6238):1036-9. Li, et al., *Nat Cell Biol*. 2015;17(10):1327-38.]. For the same reason, we prefer to keep the nomenclature as used in our manuscript. We agree with the reviewer that nomenclature clarity is important. Therefore, in the first paragraph of the Introduction section, we clearly define the nomenclature by providing the statement “Two recurrent *TERT* promoter mutations located at hotspots chr5, 1,295,228 C>T (C228T) and 1,295,250 C>T (C250T) are particularly common, which correspond to the positions 124 and 146 bp, respectively, upstream of the translation start site of *TERT*”.

Comment 2) It was also shown that mutant TERT promoters also bind NFkB proteins (although the authors of this study also incorrectly refer to this mutation as C250T). This must be mentioned in paragraph 1 of introduction.

Response: We thank the reviewer for this suggestion, but wish to remind that the present study is focused on the regulation of the mutant *TERT* promoter by the BRAF V600E/MAP kinase pathway through FOS/GABP. Many of the aspects of TERT, including the involvement of the NFkB system in the regulation of C250T, do not fit the context and theme of this study and may therefore not be all referenced in this manuscript.

Comment 3) Line numbers 35 and 288 look strikingly similar and must be changed.

Response: They are similar because they state the same background of this study in the first sentence of the Abstract and the first sentence of the Discussion, respectively, which are necessary.

Comment 4) Line number 359 mentions telomere independent roles of TERT and in

this context it must be mentioned that Tert can regulate RNA dependent RNA polymerase activity Maida et al., Nature 2009 and tRNA synthesis (Khattar et al., JCI 2016). It is also interesting that genetic evidence shows that TERT regulates myc dependent oncogeneis independent of its telomerase activity Koh et al JCI 2015 and this in my view is the first in vivo evidence and genetic evidence of a non-canonical role of TERT.

Response: Again, we wish to emphasize that this manuscript is not a comprehensive review on TERT but a focused research report on a novel regulatory mechanism of the mutant TERT promoter by the BRAF V600E/MAP kinase pathway through FOS/GABPB. Therefore, not all the studies that are interesting to the reviewer but unrelated to the theme of the present study are cited in the manuscript. To collegially respect this reviewer, however, we have now included and discussed in the discussion all the studies above suggested by the reviewer.

Comment 5) A major implication of this study could be that therapy for cancers with both BRAF and TERT mutations could depend on each other. Indeed modeling genomic instability and selection pressure in murine melanoma by Kwong et al Cell Reports 2017 will be important to quote.

Response: We believe that the reference mentioned here does not fit the content and theme of the present study; it could be a great reference elsewhere in a right context.

Comment 6) Finally, in the model (Figure 7), the authors show the GABPA dimers bind on the mutant Tert promoter. But a Cancer Discovery paper by AKincilar at al 2016 showed that TERT promoter can also be bound by tetramer of GABPA from a long distance. This is very much in line with data from Steve McKnight and others that GABPA proteins tetramerise for activation. Indeed, regulation of TERT by long distance promoter looping has been demonstrated by Kim et al., Plos Biology 2016. These papers must be suitably incorporated for correctness.

Response: We took this suggestion to revise the model and cited the most relevant (references #33-35—Akincilar's and McKnight's papers) in the revised manuscript.

To Reviewer #2

Comment I would like to thank the Authors at this point for taking up most of the Reviewer's suggestions. The present study is novel enough and the revised version of the manuscript includes enough additional details strengthening the conclusions of this study. All major and minor comments were addressed convincingly or at least well explained, especially the statistics section has been improved. In addition, all experiments are well controlled and state-of-the art methods were used throughout the study. Furthermore, the data presented by Liu et al deal with a clinically extremely relevant topic, the data are solid, and therefore, I am suggesting the revised version of this manuscript for publication in Nature Communications.

Response: Once again, we thank this reviewer for the careful review and constructive critiques, which are very valuable in helping us revise and improve the manuscript.